# Substantial oxygen consumption by aerobic nitrite oxidation in oceanic oxygen minimum zones

J. M. Beman [1✉], S. M. Vargas[1], J. M. Wilson[1,2], E. Perez-Coronel[1], J. S. Karolewski[3], S. Vazquez[1], A. Yu[1], A. E. Cairo[1], M. E. White [2], I. Koester [2], L. I. Aluwihare[2] & S. D. Wankel [3]

Oceanic oxygen minimum zones (OMZs) are globally significant sites of biogeochemical cycling where microorganisms deplete dissolved oxygen (DO) to concentrations <20 μM. Amid intense competition for DO in these metabolically challenging environments, aerobic nitrite oxidation may consume significant amounts of DO and help maintain low DO concentrations, but this remains unquantified. Using parallel measurements of oxygen consumption rates and [15]N-nitrite oxidation rates applied to both water column profiles and oxygen manipulation experiments, we show that the contribution of nitrite oxidation to overall DO consumption systematically increases as DO declines below 2 μM. Nitrite oxidation can account for all DO consumption only under DO concentrations <393 nM found in and below the secondary chlorophyll maximum. These patterns are consistent across sampling stations and experiments, reflecting coupling between nitrate reduction and nitrite-oxidizing *Nitrospina* with high oxygen affinity (based on isotopic and omic data). Collectively our results demonstrate that nitrite oxidation plays a pivotal role in the maintenance and biogeochemical dynamics of OMZs.

[1] Life and Environmental Sciences, University of California, Merced, Merced, CA, USA. [2] Scripps Institution of Oceanography, University of California, San Diego, CA, USA. [3] Marine Chemistry and Geochemistry, Woods Hole Oceanographic Institution, Woods Hole, MA, USA. ✉email: jmbeman@gmail.com

Aerobic nitrite oxidation is pervasive throughout much of the oceanic water column, playing a central role in deep ocean chemoautotrophy and carbon cycling[1,2], as well as in the oceanic nitrogen (N) cycle[3,4]. Nitrite oxidation rates are typically undetectable using $^{15}NO_2^-$ isotopic tracer in the sunlit euphotic zone (EZ), peak at the base of the EZ, and then decline with depth[5]. However, nitrite oxidation rate profiles deviate from this pattern in oceanic oxygen minimum zones (OMZs) that are depleted in dissolved oxygen (DO): rapid rates have been reported despite low DO concentrations[6–9], and nitrite oxidation itself may consume DO to low levels. OMZs are ultimately generated by high sinking fluxes of organic matter combined with reduced ventilation at depth and are typically defined by DO concentrations <20 μM (ref. [10]). Regions of the tropical oceans containing no measurable DO—termed anoxic marine zones (AMZs)[11]—are further distinguished by the accumulation of nitrite to comparatively high (>1 μM) concentrations[12]. These secondary nitrite maxima (SNM) result from anaerobic nitrate reduction to nitrite under low DO[11–13]. Accumulated nitrite may be subsequently reduced via denitrification and anaerobic ammonium oxidation—resulting in gaseous N loss[14]—or, alternatively, oxidized by NOB. Nitrite is therefore rapidly produced and consumed via multiple processes in OMZs, linking the oceans' N, carbon (C), and oxygen cycles.

Despite its centrality in OMZ biogeochemical cycles, nitrite oxidation is still poorly understood in these dynamic regions of the ocean. In particular, DO can be photosynthetically produced[15] or physically introduced to OMZs[12,16,17], requiring subsequent DO consumption in order to maintain the existence of the OMZ itself[4]. High rates of nitrite oxidation within OMZs suggest that marine nitrite-oxidizing bacteria potentially contribute to DO depletion, and therefore to the ultimate generation and maintenance of OMZs[4,6–9,15,17]. However, the contribution of nitrite oxidation to DO drawdown has not been directly quantified. While nitrite oxidation has been measured in several OMZs[6–9], oxygen consumption rate (OCR) measurements in OMZs are rare[16,18,19]. Yet, simultaneous measurements are necessary to directly quantify the contribution of nitrite oxidation to oxygen consumption and to the maintenance of OMZs.

The central question addressed by our research is whether or not nitrite oxidation is a significant DO sink in OMZs, and, if so, over what range of DO concentrations does this occur? This has direct implications for our understanding of how OMZs form and expand[4,11], as well as associated consequences for C and N cycling and loss[3,14]. For example, an additional diagnostic feature of AMZs is the presence of a secondary chlorophyll maximum (SCM) where distinct ecotypes of the globally important cyanobacterium *Prochlorococcus* produce DO within low DO waters[15,20,21]. Photosynthetic DO production is nontrivial, as production rates of up to ca. 100 nM-$O_2$ day$^{-1}$ occur in the SCM —requiring corresponding DO consumption so that DO does not accumulate[15]. SCM-based DO production can also overlap with nitrite supply via nitrate reduction—which occurs at higher DO levels than previously thought, expanding the depth range over which nitrate reduction and nitrite oxidation may interact[22,23]. However, the degree to which nitrite oxidation (versus other processes) consumes DO in the SCM remains poorly constrained[15]. Greater DO consumption via nitrite oxidation implies that less DO is used to respire organic C, while the oxidation of nitrite precludes its further reduction to gaseous forms of N[3,14].

Our understanding of how nitrite-oxidizing bacteria (NOB) oxidize nitrite in OMZs and other low-DO habitats is also evolving. NOB gain energy by oxidizing nitrite to nitrate while using DO as a terminal electron acceptor. NOB appear to have a high affinity for DO ($K_m$ values of 778–900 nM DO or lower), allowing

them to aerobically oxidize nitrite even at nanomolar concentrations of DO[22,24]. Some NOB are also metabolically flexible[25], and perhaps able to survive anaerobically through the use of alternative electron donors and acceptors[24,26,27]. Adaptations to low DO are evident in different NOB, and both *Nitrospina* and *Nitrococcus* may contribute to oceanic nitrite oxidation to varying degrees[26,28]. But although in situ nitrite oxidation rates are typically highest (>100 nmol L$^{-1}$ day$^{-1}$) under DO concentrations <11 μM (refs. [6–9]), experimental oxygen manipulations show the opposite pattern: declining rates with declining DO[22]. This may be explained by variations in genomic content among different NOB that allow them to occupy distinct, highly resolved DO niches throughout the water column[4,7,22,24,26]— including those that occur from the edge (20 μM) to the core (<5 nM DO) of OMZs. High DO affinity among some NOB also implies sensitivity to any DO introduced to incubations, and may partly explain some of the elevated rates reported in OMZs. However, DO is rarely measured directly during nitrite oxidation rate measurements, limiting our understanding of NOB affinity for DO and their activity within OMZs. These details are essential to examine in the context of ocean deoxygenation, as seemingly small variations in DO may have significant biogeochemical implications if different ecotypes within different functional groups (such as nitrite oxidizers, nitrate reducers, and organisms respiring organic matter aerobically) are adapted to specific DO concentration ranges, and so display varying sensitivities to DO.

Here, we address these open questions through parallel measurements of nitrite oxidation and overall OCR in the oceans' largest OMZ, the eastern tropical North Pacific Ocean (ETNP). Nitrite oxidation rate measurements (using $^{15}NO_2^-$), OCR measurements (using optical sensor spots), and ammonia oxidation rate measurements (using $^{15}NH_4^+$) were made along depth profiles at six stations in the ETNP, including three OMZ stations and three AMZ stations (Fig. 1). At each station, we sampled in the upper 100 m to capture the primary nitrite maximum and an expected peak in nitrite oxidation rates at the base of the EZ. We then sampled a range of DO values (from 200 μM to the anoxic SNM at AMZ stations) to quantify rate variations in response to vertical gradients in DO. How nitrite oxidation varies from the OMZ edge to its anoxic core is critical to our understanding of ocean biogeochemical cycles; as a result, we conducted oxygen manipulation experiments and examined the response of nitrite oxidation and OCR to changes in DO. Combined with natural abundance stable isotope data, 16S ribosomal RNA (rRNA) sequencing, and metagenome sequencing, we show that nitrite oxidation is a substantial— and occasionally even dominant—sink for oxygen in the ETNP.

## Results

**Nitrite oxidation rates in the ETNP.** We sampled six stations in the ETNP OMZ with DO concentrations <20 μM at depth (Figs. 1 and 2). Three stations extending out from the coast of Mexico (Stations 1–3) were AMZ stations with distinct SCMs and accumulations of nitrite in SNMs indicative of anaerobic N cycling (Fig. 2). We expected that these would be hotspots of nitrite oxidation given oxygen supply via photosynthesis in the SCM overlapping with nitrite supply via nitrate reduction. Across all stations and depths, we observed the highest nitrite oxidation rates at the AMZ stations (Fig. 2C, G, K). Station 1 is located nearest the coast and had a shallow OMZ at the time of our sampling, with 20 μM DO at 28 m depth and nitrite >1 μM at 100 m (Fig. 2A). Chlorophyll concentrations were also high in the upper water column (up to 5 mg m$^{-3}$ at 20 m), with an SCM spanning 70–125 m (Fig. 2B). Nitrite oxidation displayed a local maximum at the base of the EZ at Station 1 (20–30 m), and then

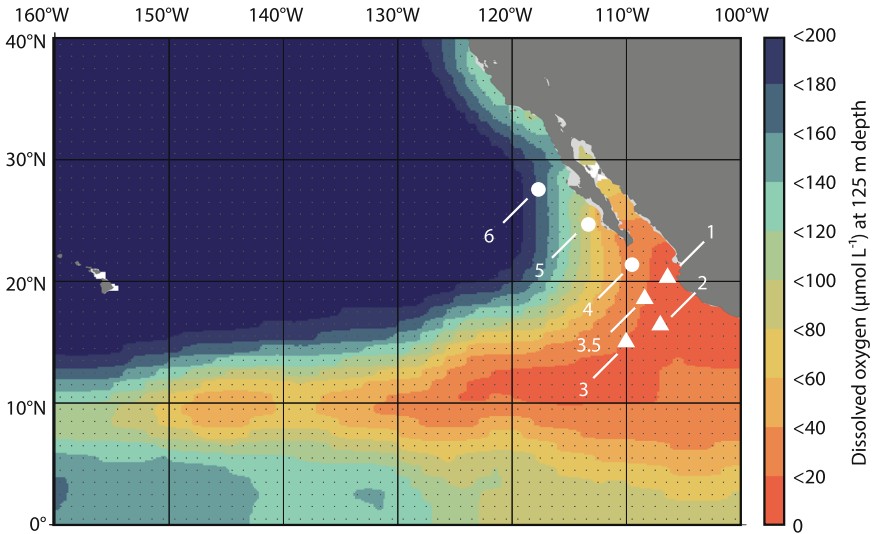

**Fig. 1 Sampling locations.** Sampling stations (numbered symbols) are plotted on dissolved oxygen concentrations (in μM) at 125 m depth from the World Ocean Atlas. Triangles denote anoxic marine zone (AMZ) Stations 1–3.5 and circles denote oxygen minimum zone (OMZ) Stations 4–6. The color scale shows dissolved oxygen concentrations in 20 μM intervals.

increased to higher levels (>100 nmol L$^{-1}$ day$^{-1}$; Fig. 2C). This increase at 100 and 125 m corresponded with the overlap between the bottom of the SCM and the top of the SNM. Nitrite oxidation rates then reached higher values at 150 m within the SNM at Station 1. Stations 2 and 3 displayed similar nitrite oxidation rate profiles to each other, including elevated rates in the SCM (Fig. 2G, K). Nitrite oxidation rates were similar in magnitude, and peak values at the base of the EZ and in the OMZ were also similar (69–96 nmol L$^{-1}$ day$^{-1}$). Depth patterns tracked oceanographic differences across the three AMZ stations, as the depth of all features increased moving offshore from Stations 1 to 2 to 3. For example, the SCM extended from 105 to 155 m at Station 2, while nitrite concentrations began to increase below 100 m; nitrite oxidation rates were elevated at 140 m and declined slightly with increasing depth (Fig. 2E–G). At Station 3, the SCM (120–180 m) and SNM (>140 m) depths were deeper, and nitrite oxidation rates increased from 100 to 200 m (Fig. 2I–K).

In contrast to these three AMZ stations (Stations 1–3), rate profiles at Stations 4–6 showed peaks at the base of the EZ followed by decreases with depth and lacked a pronounced rate increase within the OMZ (Supplementary Fig. 1). Parallel measurements of ammonia oxidation rates also showed this type of pattern at all stations (Supplementary Fig. 1). Subsurface maxima in ammonia oxidation tracked variations in the EZ across all six stations, but rates were not elevated in OMZ/AMZ waters—again contrasting with nitrite oxidation rate profiles at the AMZ stations. These data accord with earlier work in OMZs showing contrasting ammonia and nitrite oxidation rate profiles, and particularly high rates of nitrite oxidation in OMZ waters[6–8,29–31].

Initial DO concentrations for these measurements closely matched in situ values above the SCM (where DO concentrations are higher), and starting DO ranged from 260–1500 nM for measurements in and below the SCM. These DO concentrations are generally lower than those used for previous nitrite oxidation rate measurements in OMZs[6,9], but similar to work examining the oxygen affinity of nitrite oxidation[22] and overall oxygen consumption[16,19]. Elevated nitrite oxidation in the limited number of samples ($n = 5$) collected below the SCM (>125 m at Station 1, >155 m at Station 2, and >180 m at Station 3)—where little to no DO is typically available—should be considered potential rates and could have a number of possible explanations

discussed below. Within the SCM, our data support the idea that nitrite oxidation contributes to 'cryptic' oxygen cycling[15]—i.e., that DO produced via oxygenic photosynthesis is rapidly consumed.

**Oxygen consumption via nitrite oxidation.** We determined the contribution of nitrite oxidation to overall oxygen consumption via parallel measurements of OCRs using in situ optical sensor spots—which are noninvasive, provide nearly identical results as other low-level measurement approaches[32], are the only effective means of achieving substantial replication, and for which sensitivity increases as DO decreases[32,33]. Decreases in DO were measured in both nitrite and ammonia oxidation rate sample bottles, as well as in three additional replicates, to leverage statistical power for increased sensitivity to low-level DO consumption (see "Methods"). Water column OCR profiles at all stations showed exponential declines with depth and decreasing DO concentrations (Fig. 2D, H, L and Supplementary Fig. 1). Rates were highest in the upper water column and declined sharply within the upper portion of the OMZ above the SCM. The majority of measurements within the SCM—where DO may be produced via photosynthesis—were 100 s of nmol L$^{-1}$ day$^{-1}$, with an overall range of 160–1380 nmol L$^{-1}$ day$^{-1}$. Below the SCM, DO would be available more rarely (e.g., ref. [16]), and OCR measurements represent potential rates should oxygen be supplied; OCR ranged from 120 to 390 nmol L$^{-1}$ day$^{-1}$. OCR also tracked variations in DO across stations, with progressively steeper declines in OCR with depth from Station 6 to Station 1.

These OCR results are similar to the limited previous measurements that have been conducted in OMZ regions, with some key differences. In particular, they are consistent with previous measurements of rapid DO consumption in the SCM, with OCR rates ranging from 482 to 1520 nmol-O$_2$ L$^{-1}$ day$^{-1}$ in the ETSP, and from 55 to 418 nmol-O$_2$ L$^{-1}$ day$^{-1}$ in the ETNP[15]. Earlier OCR measurements conducted in the ETNP near Stations 1 and 3 (across a wide range of DO values) likewise ranged from 420 to 828 nmol L$^{-1}$ day$^{-1}$ in the SCM near Station 1, and from 101 to 269 nmol L$^{-1}$ day$^{-1}$ in the SCM near Station 3 (ref. [16]). Above the SCM, previous OCR measurements in the ETNP spanned 2260 to 662 nmol L$^{-1}$ day$^{-1}$ from the EZ to the edge of the OMZ; these values are lower than our measurements

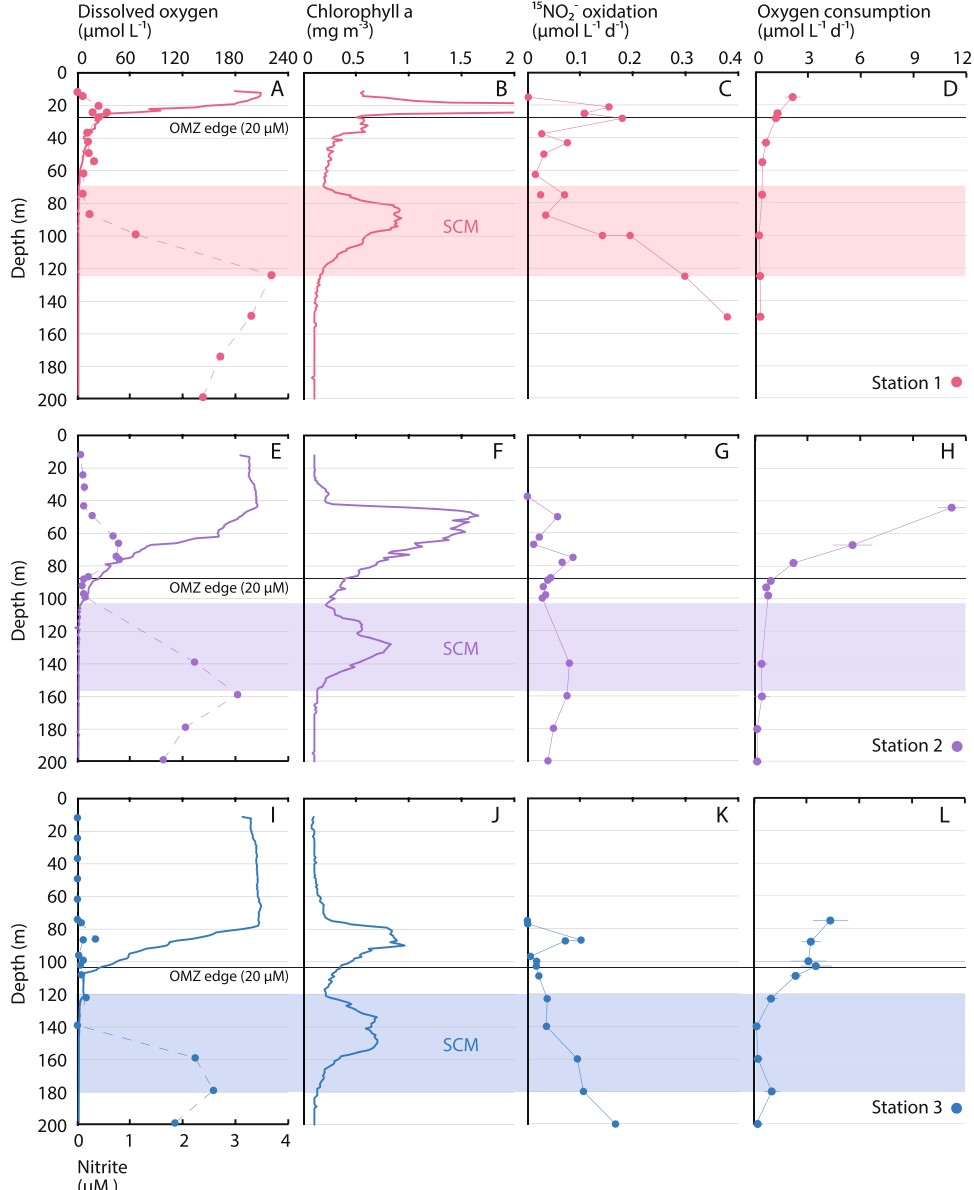

**Fig. 2 Biogeochemical depth profiles.** Profiles of **A**, **E**, **I** dissolved oxygen (solid lines) and nitrite (data points connected by dashed lines), **B**, **F**, **J** chlorophyll *a*, **C**, **G**, **K** nitrite oxidation rates, and **D**, **H**, **L** oxygen consumption rates (OCR; data presented as mean values of five independent replicates ±1 SD) show consistent variation across **A–D** Station 1, **E–H** Station 2, and **I–L** Station 3 (denoted by different colors). Black horizontal lines denote the depth of the oxygen minimum zone (OMZ), and shaded areas show the secondary chlorophyll maximum (SCM) at each station. Rates measured below the SCM should be considered potential rates (see main text). Maximum chlorophyll values at Station 1 plot off-axis.

at 44 and 67 m depth at Station 2, but in line with our remaining measurements above the SCM. OCR reached 1610 nmol L$^{-1}$ day$^{-1}$ in the SCM in Namibian shelf waters and 200–400 nmol L$^{-1}$ day$^{-1}$ in the SCM off Peru[18]. Kalvelage et al.[18] furthermore observed sharp decreases with depth in the ETSP, with rates declining from >1000 nmol L$^{-1}$ day$^{-1}$ above the SCM.

This pattern of declining OCR with increasing depth and decreasing DO was also evident in our dataset and contrasted with that of nitrite oxidation rates, which were notably elevated in the SCM at the AMZ stations (Fig. 2). We directly compared nitrite oxidation rates with OCR, assuming that each mole of nitrite is oxidized using ½ mole of O$_2$ (ref. [5]). We found that nitrite oxidation systematically increased as a proportion of overall OCR at lower DO levels (Fig. 3A, B). Nitrite oxidation was responsible for up to 69% of OCR at Station 1, although most values were closer to 10–40% at Stations 2 and 3 (Fig. 3A, B). In contrast,

ammonia oxidation contributed <5% of oxygen consumption in the OMZ (Supplementary Fig. 1). Overall, these data demonstrate consistent patterns in the contribution of nitrite oxidation to OCR, as the proportion of DO consumed by nitrite oxidation increased at progressively lower DO concentrations across multiple stations.

**Effects of oxygen manipulation experiments on nitrite oxidation and OCR.** We corroborated these observations by experimentally manipulating DO concentrations to measure the response of OCR, as well as $^{15}$NO$_2^-$ nitrite oxidation rates, to changing DO (Fig. 4). To examine responses of different assemblages in different depth regions, we conducted experiments on the edge of the OMZ at Stations 2–4, in the SCM at Stations 2 and 3, and in the SNM at Station 3 and an additional sampling Station 3.5 (Table 1). (We could not conduct additional

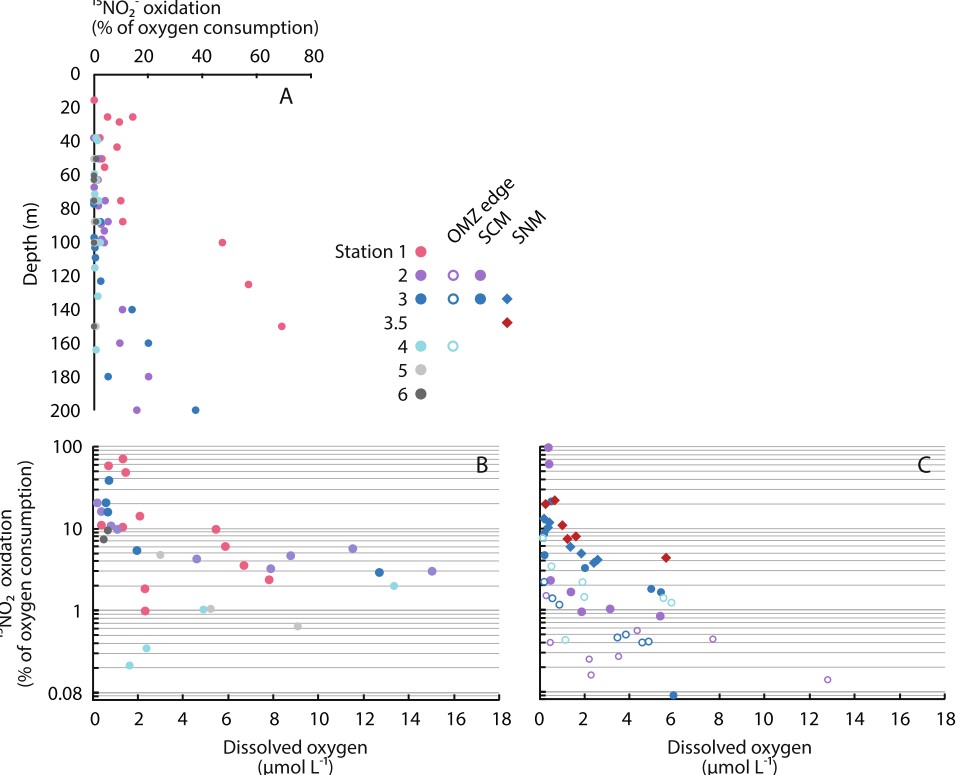

**Fig. 3 Oxygen consumption via nitrite oxidation.** The percentage contribution of nitrite oxidation to overall oxygen consumption rates (OCR) with **A** depth, and as a function of average dissolved oxygen (DO) in **B** depth profiles and **C** oxygen manipulation experiments. Colors denote different stations. Only measurements for DO < 18 μM are included in panel **B**, and all experimental bottles with dual nitrite oxidation and OCR are shown in (**C**). In panel **C**, different symbols denote experiments conducted in different regions of the water column, with two experiments conducted at Station 2 (oxygen minimum zone [OMZ] edge, open purple circles; secondary chlorophyll maximum [SCM], filled purple circles), three at Station 3 (OMZ edge, open dark blue circles; SCM, filled dark blue circles; secondary nitrite maximum [SNM], dark blue diamonds), and one each at Stations 3.5 (SNM, red diamonds) and 4 (OMZ edge, open light blue circles).

experiments at Stations 1 and 2 owing to a hurricane in the region at the time of sampling in 2018 and added Station 3.5 as a result.) Oxygen manipulations were designed to quantify OCR and nitrite oxidation across the full spectrum of DO concentrations that occur from the OMZ edge to its core—rather than solely probing their lower limits—in order to constrain rate responses to changing DO within OMZs.

Oxygen manipulation experiments were consistent with water column profiles and provide additional evidence that nitrite oxidation can be a substantial oxygen sink. In all experiments, we found that OCR declined with experimentally decreased DO (Fig. 4). These declines in OCR were particularly steep below 1–2 μM DO across different experiments. For the SCM and SNM experiments, DO concentrations >1 μM are obviously higher than expected to occur in situ (Table 1), but these concentrations were included for comparison across experiments and with earlier studies[9,15]; as a result, OCR measurements for SCM and SNM samples conducted at DO concentrations >100 s of nM represent potential rates. For OMZ edge samples, OCR values in the μM range were higher than observed in profiles—most likely due to the effects of bubbling[19], which could physically break down the organic matter present in higher concentrations at these depths (Table 1). Throughout all experiments, rate magnitudes in the 100 s of nM DO concentration range (11–820 nmol $L^{-1}$ $day^{-1}$) were similar to profile measurements (Fig. 2), as well as to previous measurements in OMZs[15,16,18,19] (see above).

DO concentrations were also continuously monitored in a subset of experimental bottles, and DO consumption was consistently linear (see "Methods"). The few exceptions occurred

in several experiments conducted at DO concentrations <235 nM; in these experiments, OCR declined as DO was consumed over time (Supplementary Fig. 2). These observations are consistent with Tiano et al.[16] and allowed us to calculate the low-level DO affinity of the community (following ref. [16]), in addition to overall values based on variations across all experimental incubations (following ref. [19]; Supplementary Note 1). However, we note that both reflect mixed assemblages of microorganisms that use DO to oxidize a variety of substrates (Supplementary Note 1). These substrates can include nitrite, as well as different forms of organic matter, reduced sulfur compounds, and possibly methane[34]. Low-level results were consistent with previous observations indicating high affinity for DO ($K_m$ ranged from 53 to 127 nM DO), and, importantly, all data demonstrate that OCR decreases as DO decreases (Supplementary Note 1 and Table 1).

In line with earlier work[22,24], nitrite oxidation rates also declined in response to decreasing DO in most of our experiments (Fig. 4). However, nitrite oxidation rates were less sensitive to declining DO than OCR (Supplementary Note 2). Nitrite oxidation was therefore responsible for progressively higher proportions of overall OCR as DO was experimentally decreased (Fig. 3C). In fact, we found that 97% of OCR in the SCM at Station 2 could be explained by measured nitrite oxidation rates—indicating that nitrite oxidation alone can sustain all DO consumption at DO concentrations just below 393 nM. However, most values were closer to 10–20%, and this contribution was obviously variable across experiments: while our experiment in the SCM at Station 2 showed that nitrite oxidation can consume all available DO (Fig. 4B), the experiment with 20 μM [DO] water at Station 2 suggests that, even at low DO

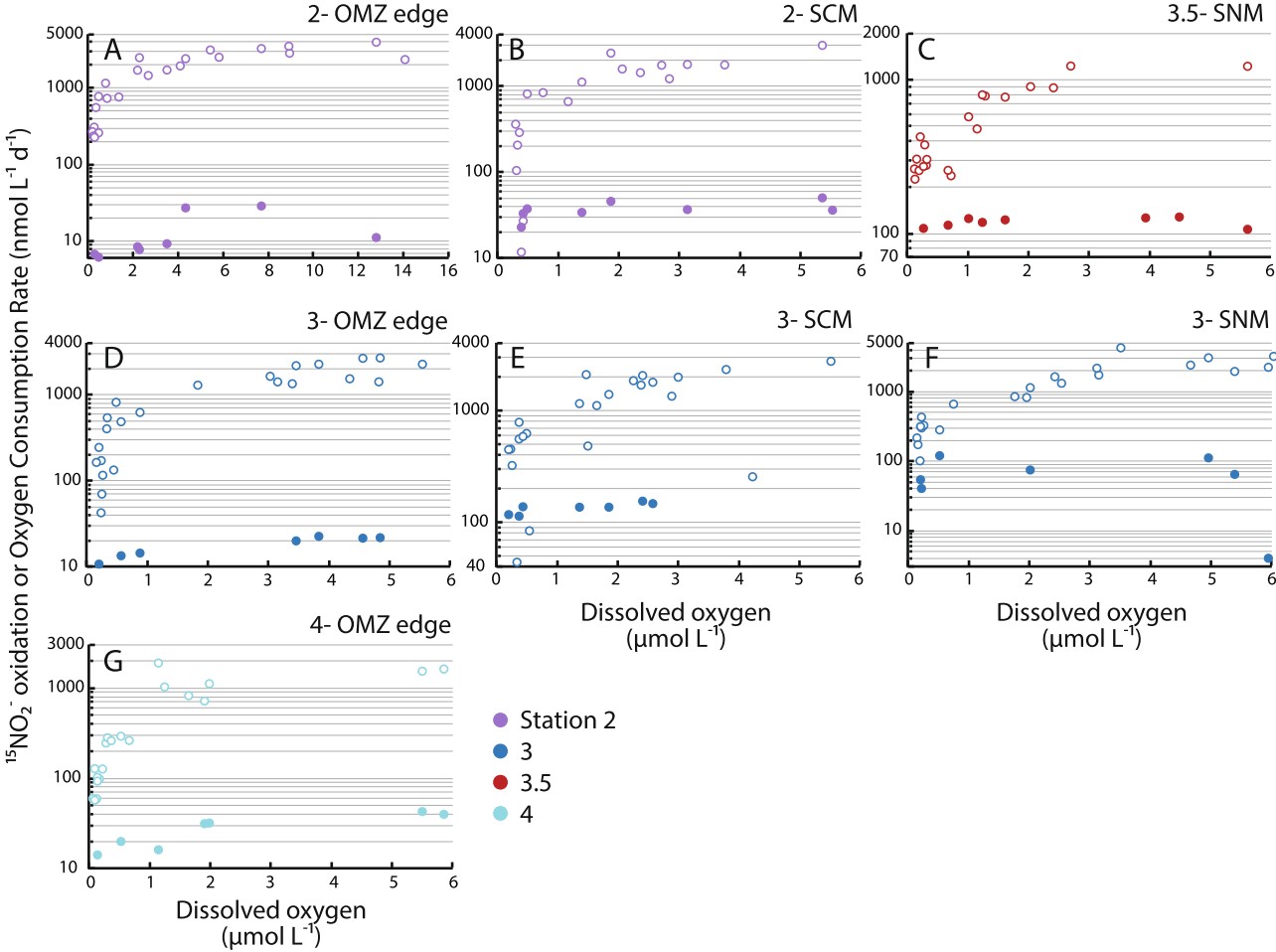

**Fig. 4 Nitrite oxidation and oxygen consumption rates (OCR) in oxygen manipulation experiments.** Nitrite oxidation rates (filled symbols) and OCR (open symbols) are displayed as a function of average dissolved (DO) in oxygen manipulation experiments. Colors denote different stations, with panels displaying data from different experiments: **A** Station 2 oxygen minimum zone (OMZ) edge (20 μM DO), **B** Station 2 secondary chlorophyll maximum (SCM), **C** Station 3.5 secondary nitrite maximum (SNM), **D** Station 3 OMZ edge, **E** Station 3 SCM, **F** Station 3 SNM, and **G** Station 4 OMZ edge. Note differences in vertical axes between experiments, and differences in the horizontal axis in panel **A** compared with the remaining panels.

levels, nitrite oxidation would not consume all DO (Fig. 4A). This highlights the fact that nitrite oxidation is unlikely to be dominant on the edge of the OMZ due to higher DO and organic matter concentrations, along with lower nitrite concentrations, compared with the SCM (Figs. 2 and 3 and Table 1). Overall, patterns in experiments were notably similar to results from rate profiles: both showed increases in the contribution of nitrite oxidation to DO consumption below 2 μM DO, both followed significant power-law relationships (experimental $r^2 = 0.40$, $P < 0.0005$; profiles $r^2 = 0.51$, $P < 0.0001$), and both displayed similar values (Fig. 3B, C).

These data tie together multiple aspects of OMZ biogeochemistry into a coherent picture. Although nitrite oxidation rates decline as DO is experimentally decreased at individual depths—and so with particular assemblages of NOB—different depths/assemblages display different properties (Fig. 4 and Supplementary Note 2). More importantly, this decline is always less severe for nitrite oxidation than for overall OCR (Figs. 2–4). NOB are therefore effective at scavenging DO (Figs. 3 and 4; ref. [15]), and the concentration range over which they become increasingly important (0.2–1 μM DO) matches model predictions involving a nitrogen–oxygen feedback loop in OMZs driven by oxygen depletion via nitrite oxidation[17].

**Isotopic, 16S rRNA, and metagenomic constraints on nitrite oxidation.** To further verify our findings, we applied natural abundance stable isotope measurements, 16S rRNA sequencing, and metagenome sequencing to water samples and nucleic acid samples collected in parallel with nitrite oxidation rate measurements (Fig. 5). The dual N and O isotopic composition of dissolved nitrate ($\delta^{15}N$ and $\delta^{18}O$, respectively; Supplementary Fig. 3) provides an effective constraint on nitrite oxidation under low DO conditions due to isotopic "overprinting" by nitrite oxidation[35,36]. While the respiratory reduction of nitrate (denitrification) leads to equal 1:1 increases in $\delta^{15}N$ and $\delta^{18}O$ (refs. [37,38]), deviations in 1:1 N:O isotope behavior arise from a decoupling of the N and O systems. These deviations are widely interpreted as reflecting cryptic reoxidation of nitrite under low DO, where the reduction of nitrate to nitrite removes an O atom, and the subsequent reoxidation of nitrite to nitrate appends a new O atom derived from ambient water[39,40]. Deviations from the 1:1 relationship are represented via $\Delta(15,18)$ values[41], where more negative values represent larger departures from the 1:1 relationship.

At Stations 1–3, dual nitrate isotope values exhibited nonlinear trends (Fig. 5B and Supplementary Fig. 4) that are consistent with isotopic overprinting by nitrite oxidation[35,36]. $\Delta(15,18)$ values were lowest (i.e., deviations were largest) at 50–150 m at Station 1, 100–140 m at Station 2, and 110–160 m at Station 3. The lower portions of these depth ranges overlapped with the upper portion of the SCM (75–12, 105–155, and 120–180 m, respectively) and

**Table 1 Details of oxygen manipulation experiments, including station, sampling depth, region of the water column (edge of the OMZ, SCM, or SNM), and experiment length; initial DO concentration measured using oxygen sensor spots (FireSting, Pyroscience) prior to experimental manipulation, TOC, and nitrite concentrations; and calculated parameters for OCRs (including both low-level and overall $K_m$ values) and nitrite oxidation rates ($K_m$ values are for dissolved oxygen in all cases, with ±standard error shown).**

| Experiment | 2-1 | 2-2 | 3-1 | 3-2 | 3-3 | 3.5-1 | 4-1 |
|---|---|---|---|---|---|---|---|
| Station | 2 | 2 | 3 | 3 | 3 | 3.5 | 4 |
| Depth (m) | 90 | 140 | 130 | 170 | 110 | 80 | 100 |
| Region of the water column | OMZ edge | SCM | SCM | SNM | OMZ edge | SNM | OMZ edge |
| Length (h) | 22 | 19 | 16 | 16 | 22 | 17 | 24 |
| Initial DO (nM) | 19,940 | 480 | 293 | 260 | 12,800 | 274 | 3480 |
| TOC (nM) | 57,965 | 45,318 | 49,118 | 46,929 | 51,211 | 44,339 | 47,397[a] |
| Nitrite (nM) | 203 | 205 | 100 | 2410 | 73 | 2650 | 68 |
| **OCR** | | | | | | | |
| Low-level $K_m$ (nM) | NA | 124 | 127 | 81 | NA | 53–104 | 82 |
| Overall $K_m$ (nM) | 3262 ± 930 | 1787 ± 689 | 1804 ± 117 | NS | NS | 1621 ± 546 | 1738 ± 755 |
| $v_{max}$ (nmol L$^{-1}$ day$^{-1}$) | 4164 ± 457 | 2757 ± 1146 | 2765 ± 788 | 5973 ± 2217 | 4025 ± 1247 | 1612 ± 262 | 2075 ± 415 |
| $r^2$ | 0.878 | 0.811 | 0.569 | 0.757 | 0.868 | 0.846 | 0.776 |
| **Nitrite oxidation** | | | | | | | |
| Overall $K_m$ (nM) | NS | 185 ± 88 | 63.8 ± 24.2 | NS | 346 ± 95 | 34.1 ± 15.0 | 1015 ± 614 |
| $v_{max}$ (nmol L$^{-1}$ day$^{-1}$) | NS | 43.9 ± 4.2 | 149 ± 6 | 71.2 ± 21.8 | 23.0 ± 1.2 | 122 ± 4 | 47.5 ± 8.6 |
| $r^2$ | 0.347 | 0.445 | 0.665 | 0.019 | 0.910 | 0.269 | 0.825 |

*OMZ* oxygen minimum zone, *SCM* secondary chlorophyll maximum, *SNM* secondary nitrite maximum, *DO* dissolved oxygen, *TOC* total organic carbon, *OCR* oxygen consumption rate, *NA* not applicable, *NS* parameter estimate not significant.
[a]Station 4 TOC sample was collected at 90 m.

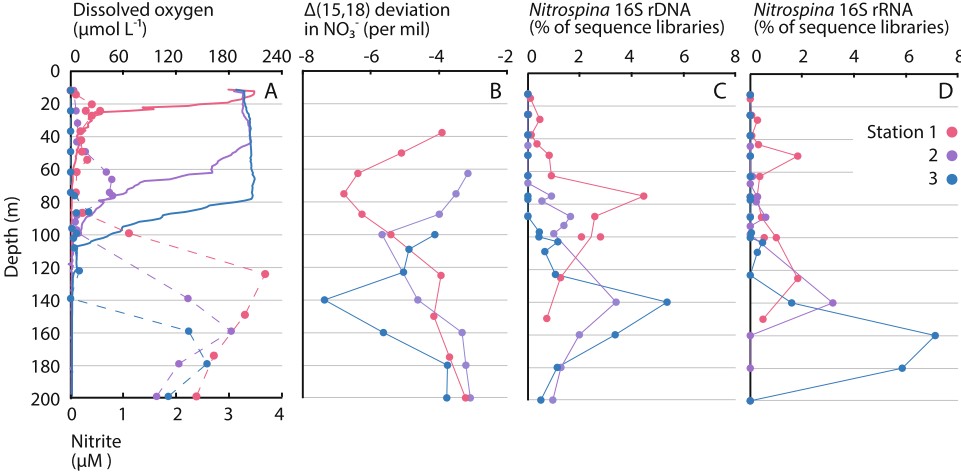

**Fig. 5 Nitrate isotopic composition and *Nitrospina* bacteria.** Depth profiles of **A** dissolved oxygen (solid lines) and nitrite (data points connected by dashed lines), **B** Δ(15,18) dual-isotope deviations in nitrate, and *Nitrospina* amplicon sequence variants (ASVs) as a percentage of **C** 16S rDNA sequence libraries and **D** 16S rRNA sequence libraries. Colors denote different sampling stations.

tracked depth variations between stations. Peak deviations at Stations 1 and 2 were just above the SCM (75 and 100 m), while the peak at Station 3 corresponded closely with the SCM (140 m). Throughout these ETNP OMZ sites, our nitrate isotope data are consistent with previous observations and interpretations of rapid recycling between nitrate and nitrite[35,41,42], ultimately evidenced by isotopic overprinting of the nitrate reduction signal by nitrite oxidation.

16S rRNA gene and transcript sequencing revealed a similar pattern to rate profiles, experiments, and isotopic data, while also enabling identification of NOB that are abundant and active in the ETNP OMZ. Based on DNA, *Nitrospina* 16S rRNA amplicon sequence variants (ASVs; ref. [43]) comprised up to 3.4–5.4% of all ASVs at Stations 1–3 (Fig. 5C), and relative abundances were consistent with biogeochemical data. Abundances and isotopic deviations were well-correlated ($r^2 = 0.70$–$0.84$, $P < 0.05$) at Stations 1 and 3, for instance, while abundances correlated with nitrite oxidation rates ($r^2 = 0.20$–$0.46$, $P < 0.05$) at Stations 2 and 3. Patterns observed for DNA samples were accentuated for RNA,

with discrete peaks in 16S rRNA transcripts generally occurring at deeper depths (Fig. 5D). *Nitrospina* 16S rRNA peaked at 125 m at Station 1 (1.8% of all 16S rRNA transcripts; with an additional upper water column peak at 50 m), 140 m at Station 2 (3.2%), and particularly 140–180 m at Station 3 (1.6–7.1%). All of these depths lie within the SCM at each station. Although 20 *Nitrospina* ASVs were identified, three were dominant (each >1% of all ASVs, together constituting 72–100% of all *Nitrospina* 16S rRNA gene sequences and transcripts). One of these ASVs was found only in OMZ samples at Stations 1–3 and was not detected above the OMZ, nor at Stations 4 and 5 (Supplementary Fig. 5). This was also the lone ASV observed in the OMZ at Station 3, where *Nitrospina* were most active based on their comparatively high percentages of all 16S rRNA transcripts. In contrast, no *Nitrococcus* ASVs and only one low-abundance *Nitrospira* ASV were identified out of >11,000 ASVs and >3.5 million 16S sequences from 73 DNA and 73 RNA samples. In line with earlier work in the ETNP[7,26], other OMZs[6,44], and the variations in DO affinity observed across oxygen manipulation experiments (Fig. 4),

**Table 2 Relative abundance of *Nitrospina* reads, functional genes from *Nitrospina*, and *Prochlorococcus* reads (all expressed per million reads) within metagenomes collected on the edge of the oxygen minimum zone (OMZ), in the secondary chlorophyll maximum (SCM), and in the secondary nitrite maximum (SNM) at Stations 1–3.**

| Station | Depth (m) | Depth region | Initial DO (μM) | *Nitrospina* (all reads) | *Nitrospina* nitrite oxidoreductase (*nxr*) | *Nitrospina* cytochrome *c* oxidase | *Nitrospina* cytochrome *bd* oxidase | *Prochlorococcus* (all reads) |
|---|---|---|---|---|---|---|---|---|
| 1 | 25 | OMZ edge | 19.9 | 1084 | 2 | 0 | 0 | 865 |
| 1 | 87.5 | SCM | 0.46 | 31,920 | 65 | 20 | 3 | 47,992 |
| 1 | 100 | SNM | 0.30 | 21,196 | 37 | 22 | 2 | 10,765 |
| 2 | 89 | OMZ edge | 15.3 | 22,442 | 38 | 12 | 10 | 57,194 |
| 2 | 130 | SCM | 0.78 | 33,290 | 57 | 20 | 4 | 33,522 |
| 2 | 160 | SNM | 0.60 | 21,725 | 37 | 28 | 2 | 4157 |
| 3 | 123 | OMZ edge | 13.0 | 18,647 | 34 | 10 | 2 | 54,709 |
| 3 | 140 | SCM | 0.74 | 34,227 | 55 | 18 | 4 | 37,202 |
| 3 | 180 | SNM | 0.45 | 17,617 | 30 | 26 | 2 | 4881 |

Initial dissolved oxygen (DO) concentrations were measured in unmanipulated incubation bottles using oxygen sensor spots (Fibox, Loligo Systems) immediately following sample collection.

these results indicate that particular *Nitrospina* ecotypes—and one ASV in particular—are significant for low-oxygen nitrite oxidation, while other ASVs may be more important at different depths and DO concentrations.

To validate the genetic potential for nitrite oxidation by *Nitrospina* under low DO concentrations, we sequenced metagenomes collected on the OMZ edge, at the SCM, and within the OMZ core at Stations 1, 2, and 3. Nitrite oxidoreductase (*nxr*) genes from *Nitrospina* were prevalent at all stations and depths, establishing the genomic potential for nitrite oxidation throughout the ETNP OMZ (Table 2). *nxr* and all *Nitrospina* genes were most abundant in the SCM, consistent with a range of omic data showing maximal *Nitrospina* gene or enzyme abundances in the upper portion of AMZs[2,7,15,26]. In addition, multiple high-affinity cytochrome *c* oxidase genes from *Nitrospina* were present in all samples—indicating that *Nitrospina* is capable of consuming DO at the low concentrations found within the OMZ. These genes were more common in SCM and SNM metagenomes than at the OMZ edge, consistent with *Nitrospina* ecotype distributions, and with the lower DO concentrations found in the SCM and SNM (Table 2). Cytochrome *bd*-type oxidase genes from *Nitrospina* were also detected in all metagenomes except the OMZ edge sample at Station 1—although these lack quinol binding sites and may not function as canonical oxidases[26,45]. Sun et al.[26] also suggested that chlorite dismutase genes may be relevant for anaerobic metabolism in *Nitrospina*, and these were present in all metagenomes (Supplementary Table 1). We also recovered formate dehydrogenase and nitrate reductase genes indicative of anaerobic metabolism in *Nitrospina*[25] (Supplementary Table 1). Both of these genes were absent from OMZ edge metagenomes at Stations 1 and 2, showing higher relative abundances in the SCM, and especially the SNM. *Nitrospina*-derived nitrate reductase genes were present at similar levels as nitrite oxidoreductase in the SNM, while less common in the SCM (Supplementary Table 1). Finally, *Prochlorococcus* genes were prevalent from the OMZ edge to core, and especially within the SCM, supporting the idea that oxygenic photosynthesis and cryptic oxygen cycling occur throughout the ETNP OMZ (Table 2). Overall, metagenomic data are consistent with experimental data and 16S data and indicate that *Nitrospina* is tightly tuned to DO concentrations in the ETNP.

## Discussion

Assembled together, our results provide a comprehensive view of nitrite oxidation and its contribution to oxygen consumption in the ocean's largest OMZ. Rate measurements, oxygen

manipulation experiments, stable isotopic data, and molecular data all converge on the fact that nitrite oxidation is active from the base of the EZ into the OMZ. Nitrite oxidation was particularly significant within the SCM at Stations 1–3: nitrite oxidation rates were elevated (Fig. 2), isotopic overprinting of nitrate by nitrite oxidation was evident (Fig. 5B and Supplementary Fig. 4), and *Nitrospina* were abundant and active (Fig. 5C, D). All of these data tracked systematic progression in the depth of the SCM across stations (Fig. 2), and indicate that nitrite oxidation is important for DO utilization in the SCM. Based on rate profiles, nitrite oxidation ranged from 10–47% of OCR in the SCM (Fig. 3B), while experiments indicate maximum percentages of 13–97% (Fig. 3C). Stations 4–6—which lack an SCM and SNM—provide a notable contrast, as do ammonia oxidation rate profiles (Supplementary Fig. 1).

Peaks in the magnitude of departure from a 1:1 nitrate reduction signal (e.g., isotopic overprinting by nitrite oxidation), and in *Nitrospina* ASV abundances, further indicate that nitrite oxidation is active above and below the SCM (Fig. 5). At Stations 2 and 3, nitrite oxidation rates above the SCM were similar to rates deeper in the water column, while Station 1 showed a rate increase within the AMZ (Fig. 2). In OMZs/AMZs, DO may be photosynthetically produced in the SCM[15,46] or introduced through physical mixing[12,16,46], but both of these DO sources would be less significant below the SCM. Instead, the introduction of DO during sample manipulation effectively measures the potential for oxygen consumption in samples collected below the SCM. Short of conducting in situ incubations, such DO contamination is essentially unavoidable and is the reason that we also performed oxygen manipulation experiments (Fig. 4)—which also have limitations[19]. In line with earlier work in OMZs facing these same challenges[16,18], oxygen consumption potential was significant even below the SCM. Most importantly, the inclusion of DO sensor spots in all bottles allowed us to directly measure DO in every incubation, and therefore to quantify nitrite oxidation and OCR as a function of DO across all profiles and experiments. Patterns in OCR and nitrite oxidation were notably consistent between profiles and experiments (Figs. 3B, C). This similarity indicates that—independent of methodology—nitrite oxidation increases in relative importance below 2 μM DO.

To make our findings widely relevant to current and emerging OMZ regions, we examined the full range of DO concentrations found in OMZs, rather than in the AMZ exclusively. However, consistent with isotopic data, we found that *Nitrospina* 16S rRNA transcripts increased in relative abundance within the AMZ (Fig. 5). RNA samples were collected from the CTD rosette and rapidly filtered, so if these data capture a response to oxygen exposure, this

response is exceptionally rapid and consistent. Alternatively, these data indicate that *Nitrospina* bacteria are active within the AMZ. One possibility is that *Nitrospina* bacteria survive anaerobically for periods of time until provided DO[25]. Oxygen-evolving mechanisms from nitrite have also been proposed for anaerobic methane-oxidizing bacteria[47] that are active in OMZs[48]. The presence of *Prochlorococcus* below the SCM may allow low-level DO production via oxygenic photosynthesis (Table 2). Finally, multiple previous studies have detected DO at ca. 500 m depth within the core of the ETNP AMZ[16,46]. This is substantially deeper than our experiments and well below the SCM (Fig. 2)—suggesting that DO can be periodically introduced even deep within the AMZ. Regardless of the mechanism, isotopic data and 16S rRNA data suggest that transient nitrite oxidation coupled to nitrate reduction occurs below the SCM[17].

However, our primary focus was on oxygen consumption via nitrite oxidation and its implications for the formation, maintenance, and expansion of OMZs. Two clear and remarkably consistent patterns emerged from both water column profiles and oxygen manipulation experiments: (i) overall OCR declined with decreasing DO in profiles and experiments (Figs. 2C and 4), while (ii) nitrite oxidation increased as a proportion of OCR as DO declined in both profiles and experiments (Fig. 3B, C). Nitrite oxidation was significant throughout the OMZ and experiments, but increased in relative importance below 2 µM DO—typically ranging from 10 to 40% of OCR—and was only dominant (>50% OCR) at low DO concentrations (Table 1). For example, nitrite oxidation consumed nearly all DO produced in the SCM experiment at Station 2 (Fig. 3C). Variation in the proportion of overall OCR attributable to nitrite oxidation could reflect fluctuations through time[17], as well as differences in substrate availabilities and affinities relative to DO for different processes consuming DO. Other than nitrite oxidation, aerobic respiration of organic matter is most significant for OCR given relative substrate concentrations (organic C > methane or reduced sulfur compounds; refs. [48–50]), as well as high affinities for DO among heterotrophic bacteria[51–53]. Our data indicate that aerobic heterotrophic respiration is likely the dominant DO-consuming process from the edge of the OMZ (20 µM DO) until OCR declines precipitously at lower DO concentrations (Fig. 4 and Table 1).

Many aerobic heterotrophs are also facultatively anaerobic, switching to nitrate reduction under low DO[13,54]. Along with a substrate-driven decline in aerobic respiration rates with decreasing DO, this shift to nitrate reduction likely explains the steep declines in OCR observed in our experimental data (Fig. 4). Production of nitrite from nitrate reduction also provides a needed substrate for nitrite oxidation, and is consistent with isotopic data (Fig. 5B and Supplementary Fig. 4). We suggest that such cryptic nitrite/nitrate cycling explains the adaptations of particular *Nitrospina* bacteria to AMZs: although these are low-DO environments, they are also high-nitrite environments, representing a trade-off in the availability of electron donor and acceptor. The fact that nitrite oxidation rates increase with increasing nitrite into the micromolar range[9], and that nitrate is reduced at similar rates[55], provides additional support for this idea. Our results further indicate that nitrite oxidation is carried out by just a few *Nitrospina* ASVs under these conditions (Supplementary Fig. 5)—in contrast with the diversity of other microbial groups that may be involved in overall oxygen consumption[4,56]. These *Nitrospina* are disproportionately biogeochemically important given their roles in DO consumption and regeneration of nitrate, which precludes its further reduction. For these organisms, the costs of life at low DO are offset by the advantages of persistently elevated nitrite concentrations.

At the same time, efficient scavenging of DO by *Nitrospina* can maintain DO at low levels favorable for nitrate reduction[17,57].

Rate profiles, oxygen manipulation experiments, natural abundance measurements, 16S rRNA sequencing, and metagenomic data all favor this idea: the proportion of DO consumed by nitrite oxidation increased at progressively lower DO concentrations in both profiles and experiments (Fig. 3B, C); isotopic anomalies are indicative of coupled nitrite oxidation and nitrate reduction (Fig. 5B and Supplementary Fig. 4); and the low-oxygen adaptations of particular *Nitrospina* support high abundances and activity (Table 2, Fig. 5C, D, and Supplementary Fig. 5). By placing quantitative bounds on the contribution of nitrite oxidation to DO consumption, we resolve the niche of nitrite-oxidizing *Nitrospina* within OMZs, constrain their contributions to coupled oxygen and nitrogen cycling, and convincingly establish that nitrite oxidation is pivotal in the maintenance and biogeochemical dynamics of these important regions of the ocean.

## Methods

**Sample collection.** Samples were collected in April 2017 and June 2018 aboard the R/V *Oceanus*, with samples collected in Mexican territorial waters under Instituto Nacional de Estadística y Geografía permits EG0062017 and EG0032018, and Permiso de Pesca de Fomento permits PPFE/DGOPA-016/17 and PPFE/DGOPA-027/18. At each station, conductivity/salinity, temperature, depth, pressure, chlorophyll fluorescence, and photosynthetically active radiation (PAR) were measured by a SeaBird SBE-9plus CTD, SBE-3F temperature sensor, SBE-43 DO sensor, WetLabs ECO-FLR Fluorometer, and Biospherical QCP2200 PAR sensor. Initial casts were used to measure DO and nutrient profiles to guide subsequent sampling. Nutrient samples were analyzed for $NH_4^+$ and $NO_2^-$ aboard the ship, with additional shore-based analyses of combined $NO_3^- + NO_2^-$ and $PO_4^{3-}$ at the University of California Santa Barbara Marine Science Institute Analytical Lab (Supplementary Materials and methods).

At all stations, water samples were collected in the upper 100 m to capture variation in the upper water column, and we then sampled across a range of DO levels and nitrite levels to capture variation in the OMZ (Supplementary Table 2). Samples were collected at 200, 100, 50, 20, 10, 5, and 1 µM [DO] at all stations. Because the OMZ is shallower and more intense moving south and nearshore in the ETNP (Fig. 1), these DO levels can occur within the upper 100 m, such that upper 100 m and DO-based sampling overlapped. This overlap was greater at Station 1, followed by 2 and 3. At these AMZ stations, samples were also collected at three depths below the 1 µM [DO] level of the CTD, within the SNM (Supplementary Table 2). Experiments were conducted for 24 h in the dark in a cold van adjusted to ambient temperature, with separate incubations (and therefore temperatures) conducted for samples from the upper 100 m, for oxygen/nitrite-based sampling, and for oxygen manipulation experiments at each station.

**OCR measurements and oxygen manipulations.** Two types of sensor spots were used to measure OCR and manipulate DO, one with a wider range and detection limit of 100 nM (Fibox, Loligo Systems, Viborg, Denmark), as well as trace-level spots with a detection limit of 10 nM (FireSting, Pyroscience, Aachen, Germany). Loligo sensor spots were used in 2017 for water column profiles given the wider DO range, while FireSting trace-level spots were used in 2018 to establish oxygen manipulation experiments and measure OCR within them. Optical sensors were calibrated using DO-saturated water and sodium sulfite-saturated and He-purged water. On the upper end, Fibox sensor spot DO measurements were highly correlated with CTD DO values ($r^2 = 0.996$, slope = 0.997, $P < 0.00001$) for [DO] > 1 µM. On the lower end, we regularly checked the "0 nM" concentration with repeated measurements of sodium sulfite-saturated water and He-purged water, and our lowest values measured in experiments were below the detection limit (10 nM) of the FireSting. In both profiles and experiments, OCR was measured by the decrease in DO over the course of experiments[32,58].

Water column profiles of OCR were measured using five replicates at each depth, including one each with tracer-level (5–10% in situ concentration measured at sea) addition of $^{15}NH_4^+$ or $^{15}NO_2^-$ to measure ammonia/nitrite oxidation (see below). Loligo sensor spots were attached to the inside of 300 mL Wheaton BOD glass bottles using silicone glue. At sea, bottles were filled to at least three times overflowing via slow laminar flow, were rapidly capped and transferred to the cold van, and initial DO measurements were made. DO was measured subsequently at 10–14 h, and at the end of the 20–24 h incubations. The majority of OCR rate values calculated at these 10–14 and 20–24 h measurement time points were highly correlated with each other ($r^2 = 0.968$–0.995; slopes = 1.06–1.19; all $P < 0.0001$ across different stations), indicating that OCR did not accelerate or decrease substantially over the course of the incubations. The only exceptions were three sampling depths from station 3 (77, 87, and 97 m) showing nonlinearity, and so we report data from duplicate OCR measurements conducted at similar depths (75, 88, and 100 m) during the previous 24 h period (when nitrite oxidation rates were not measured in tandem). After incubation, endpoint DO measurements were made,

and then 50 mL samples from relevant bottles were frozen for nitrite and ammonia oxidation rate measurements.

Oxygen manipulation experiments were conducted in 500 mL serum bottles with attached FireSting sensor spots. We conducted a total of seven experiments, with experiments on the OMZ edge and in the SCM at Station 2; on the OMZ edge, in the SCM, and in the SNM at Station 3; in the SNM at Station 3.5; and on the OMZ edge at Station 4. For each experiment, a total of 24 bottles were filled with water collected at a specific depth, sealed with deoxygenated stoppers, and then bubbled with ultrapure He gas (15–30 min) while DO was monitored. We introduced an He headspace of 20 mL, and air was injected into bottles to raise DO concentrations when needed. For each experiment, we established initial DO values ranging from 10 to 1000 s of nM (Fig. 4), and eight bottles had tracer-level $^{15}NO_2^-$ additions, eight bottles had tracer-level $^{15}NH_4^+$ additions, and eight were unlabeled. For all experiments and water column profiles, bottles were cleaned with 0.1 M HCl and triple rinsed with water between incubations, and dedicated bottles were used for the different $^{15}N$ labels.

OCR was measured in all experimental bottles based on starting and ending DO values, and samples for nitrite oxidation were collected from $^{15}NO_2^-$ labeled bottles at the end of the experiments. Four to eight bottles in each experiment were also continuously monitored for DO concentration changes throughout the experiment. We tested DO consumption curves for linearity over time using the maximal information coefficient (MIC; ref. [59]) and linear regression. MIC effectively captures both linear and nonlinear relationships, such that differences between higher MIC scores and lower regression coefficients are indicative of nonlinear relationships, whereas high MIC scores combined with high $r^2$ values are indicative of strong linear relationships[59]. We found the vast majority of DO time courses were linear ($r^2 > 0.6$, MIC-$r^2 < 0.2$). The few exceptions—one incubation bottle each in the SCM at Stations 2 and 3, one in the SNM at Station 3, two in the SNM at Station 3.5, and one on the OMZ edge at Station 4—all showed a decline in OCR over time. All of these incubation bottles had an average DO < 235 nM, and were used to calculate affinity for DO at these low concentrations (Supplementary Note 1).

**Nitrate isotopic measurements.** Both natural abundance and $^{15}N$-tracer-based nitrate isotopes were quantified by use of the denitrifier method[60,61]. Samples were collected from the rosette or at the end of incubations, filtered through a 0.2 μm filter, and then frozen until analysis. In the lab, 20–30 nM of $NO_3^-$ were quantitatively converted to $N_2O$ by a culture of the denitrifying bacterium, *Pseudomonas aureofaciens*. Product $N_2O$ was cryogenically purified and trapped under the flow of ultra-high-purity He, before being introduced to an IsoPrime 100 isotope ratio mass spectrometer. Isotope ratios were normalized to international reference materials (USGS 32, USGS 34, and USGS 35) and reported using standard delta notation. Any nitrite present in the sample was removed by the addition of sulfamic acid[37]. In samples from the upper 100 m, where $NO_3^-$ concentrations were low, "carrier" $NO_3^-$ of known isotopic composition was added and sample compositions were calculated by mass balance. The precision of natural abundance isotope ratio measurements was ±0.3‰ and ±0.4‰ for $\delta^{15}N$ and $\delta^{18}O$, respectively. $\delta^{15}N$ and $\delta^{18}O$ values were used to calculate the Δ(15,18) deviation from the 1:1 line following Sigman et al.[41].

**Nitrite oxidation rates.** Nitrite oxidation rates were measured by adding 98 atom percent (at.%) $^{15}NO_2^-$ to a final concentration of 12–184 nM L$^{-1}$—representing 5–10% of in situ $NO_2^-$ concentrations (with the exception of some samples in the upper 100 m with no measurable nitrite)—and measuring the accumulation of $^{15}N$ label in the $NO_3^-$ pool following Beman et al.[7]. After incubation for ~24 h, samples were frozen at sea. Upon thawing in the laboratory, excess $^{15}NO_2^-$ was removed following Granger and Sigman[37]. In brief, sulfamic acid (~8–10 μL mL$^{-1}$) was added, the samples were shaken and allowed to sit for >5 min, and then samples were neutralized by adding NaOH (4 M, ~11 μL mL$^{-1}$) prior to analysis.

Rates of $^{15}NO_2^-$ oxidation ($^{15}R_{ox}$) were calculated using Eq. (1) from Beman et al.[7] adapted from Ward et al. (ref. [62]):

$$^{15}R_{ox} = \frac{(n_t - n_{oNO_3^-}) \times [NO_3^-]}{(n_{NO_2^-}) \times t} \tag{1}$$

where $n_t$ is the at.% $^{15}N$ in $NO_3^-$ measured at time $t$ corrected for $^{15}NO_3^-$ contamination of the $^{15}NO_2^-$ stock, $n_{oNO_3^-}$ is the measured at.% $^{15}N$ of unlabeled $NO_3^-$, $[NO_3^-]$ is the concentration of the $NO_3^-$ pool, and $n_{NO_2^-}$ is the exponential average atom% of $NO_2^-$ over time t. $n_{NO_2^-}$ was calculated by isotope mass balance based on initial $NO_2^-$ concentrations, measured isotopic composition, added $^{15}N$-labeled $NO_2^-$, and measured $^{15}NH_4^+$ oxidation rates (which produce unlabeled $NO_2^-$; ammonia oxidation rate measurements followed Beman et al.[30]).

RNA/DNA extractions, 16S rRNA sequencing and analysis, metagenome sequencing and analysis, and organic matter analysis are reported in Supplementary Materials and methods.

Statistical analyses were carried out in the R statistical environment (RStudio Version 1.0.136). Linear regressions were performed using the "lm" function in R; Michaelis–Menten curve fits (Supplementary Note 2) were performed using the "drm" function in the R package drc (ref. [63]).

**Reporting summary.** Further information on research design is available in the Nature Research Reporting Summary linked to this article.

## Data availability

The CTD data generated in this study have been deposited in the Rolling Deck to Repository, with 2017 data available under accession number OC1704A and 2018 data available under accession number OC1806A. Nutrient and biogeochemical rate data are available through the Biological and Chemical Oceanography Data Management Office under project number 863208 (https://www.bco-dmo.org/project/863208). Sequence data are available in the Sequence Read Archive, with 16S data available under BioProject PRJNA192803 (https://www.ncbi.nlm.nih.gov/bioproject/PRJNA192803) and metagenome data available under BioProject PRJNA634212 (https://www.ncbi.nlm.nih.gov/bioproject/PRJNA634212).

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

## Acknowledgements

This work was supported by NSF CAREER Grant OCE-1555375 to J.M.B. Metagenome sequencing was supported by the UCMEXUS-CONACyT Collaborative Grants Program (joint awards to J.M.B. and José García Maldonado). We thank the officers and crew of the R/V *Oceanus* for their help at sea and also thank Sarah Abboud, Daniela Alonso, DeVonyo Bills, José García Maldonado, Jorge Montiel Molina, and Kevin Testo for their assistance in the field.

## Author contributions

J.M.B. designed the study and performed the research with S.M.V, J.M.W., E.P.-C., J.S.K., S.V., A.Y., A.E.C., M.E.W., and I.K.; L.I.A. and S.D.W. contributed new analytical tools, and J.M.B. wrote the paper.

## Competing interests

The authors declare no competing interests.
