## [Peer Review File · Nature Communications]

Reviewers' comments:

Reviewer #1 (Remarks to the Author):

In the manuscript Constraining the contribution of aerobic nitrite oxidation to oceanic oxygen minimum zone formation, Beman and colleagues present a study that builds on the body of literature looking at the role of nitrite oxidation in low oxygen systems. Specifically, the work demonstrates the potential contribution of nitrite oxidation to oxygen consumption, and how this systemically increased at low ($< 2 \mu\text{M}$) oxygen levels, and thereby the large role nitrite oxidation may be playing in biogeochemical cycling within ODZs. The overarching hypothesis is noteworthy and certainly of interest to the ODZ community. However in its current format, the experiments and resulting arguments put forward seem a little disconnected from insitu conditions in places, particularly regarding the ODZ core, where oxygen is below detection and a number of other issues deserve attention to help strengthen the study, prior to publication.

In order to measure oxygen consumption rates, oxygen needed to be available within your incubations and in all cases shown in Figure 3 it was enough to support all of the nitrite oxidation observed. Is this the case insitu (particularly with regards to the ODZ core)? The starting oxygen concentrations relative to insitu concentrations in these experiments urgently needs to be touched upon and what that means for the nitrite oxidation rates presented. While I understand the value here, and excitement especially with the potential for cryptic oxygen cycling in the SCM, it needs to be highlighted clearly that your incubations do not match insitu conditions in the core and it is more about the potential. Use your incubations to more clearly link to ideas of oxygen availability, and subsequent impacts on biogeochemical cycling.

Some comments directly linked to this:

- Line 300 to 302: here you mention that in one-two of the five replicates in the profile measurements has trace amounts of oxygen, this is not clear to me. What do you mean by trace amounts? Surely you needed oxygen to be measurable to measure your oxygen consumption rates? I agree that this is unavoidable, but also why it is truly essential that you list the starting oxygen concentrations for these experiments.
- Line 315 to 318: Here you mention the potential for an anaerobic metabolism, but with how the arguments have been built to this point i.e. all nitrite oxidation seen can be aerobic, this seems a little disconnected.
- I would recommend that you try to split up your discussion of experiments that do / dont represent insitu conditions i.e. oxycline and core, I think it will make it much easier for the reader to follow and relate to other literature.

In both the introduction (line 66) and in the discussion (line 157), the authors only refer to / compare to the oxygen consumption rates by Kalvelage et al, 2015, these measurements are largely from shelf ODZs, so are not the best comparison point for your open ocean work. The work however carried out by the group of Niels-Peter Revsbech, and in particular Tiano et al (2014), which carried out oxygen consumption rates in the ETNP (one site very close to one of your own) are a great comparison and raise a number of questions. While there are some similarities there are also some striking differences, particularly with regards to half saturation constants, which Tiano has in the range 10 to 200 nM with the lowest presented here being 1621 nM. The idea of bottle effects presented by Tiano and keeping experiments short for oxygen consumption rates (< 20 hours) to avoid bottle effects, also needs to be commented upon, it is unfortunate that only start and end oxygen concentrations are available from the experiments presented in this paper (any time lags, exponential behavior etc would have been missed) – this points to the potential for your rates to be overestimates, and this needs to be discussed.

Specific comments

Title: I find the use of the word 'formation' quite strong, yes your experiments indicate that nitrite oxidizers have a potentially important role to play in oxygen consumption, but without sampling on a larger spatial and temporal scale, along with modelling, the word formation seems a bit too large of a step. Throughout, regarding the link to formation and maintenance of the ODZ, this needs to be toned down in places.

Line 51, 63, 80: throughout, language such as low, intrinsically low, trace is used (these are just a few examples), please try to be quantitative wherever possible, these words have very different meanings for

different individuals.

Line 62 to 64: references are needed at the end of this sentence.

Line 85: 'declining rates with declining DO' here and in this paragraph in general it is surprising that you don't take the time to explain in more detail the kinetics work done to date (e.g. Sun et al, 2017, Bristow et al, 2016) introducing the half saturation constants, high oxygen affinity etc, as it is this work you are attempting to build on.

Line 117: it would be beneficial for the reader to add in the depth of the oxic-anoxic interface (for all stations)

Line 134: My understanding is that your definition of an OMZ station is that oxygen drops below 20 μM , looking at Figure S1 this doesn't seem to be the case for Stations 4 and 6? Is this just due to the depth ranges plotted?

Line 275: this is the first mention of an anaerobic metabolism, this needs some context prior to this. Is there any other evidence in the metagenomes for an anaerobic metabolism? It would be great to hear more about your findings from these metagenomes, this data seems a little underutilized.

Line 318 to 319: how do the rates of oxygen production observed in Garcia-Robledo et al, 2017 align with what you have seen / expect?

Line 342: I suggest you link to the work of Sun et al, 2017 here and the K_m for nitrite.

Line 397: which experiments are you referring to when you say 'deoxygenation experiments'?

Line 402: please clarify what you mean by 'within 5nM of 0' – is the 0 you refer to here 10nM (the detection limit mentioned)?

Line 410 to 412: the initial DO concentrations in the BOD bottles, how close were these to insitu values (the detection limit of your SBE-43 will also be important here, line 374)? Were the BOD bottles cleaned prior to use to avoid contamination with organics which could lead to bottle effects (e.g. Tiano et al, 2014)

Line 415 to 423: a lot more detail is needed here to allow someone to repeat these oxygen manipulation experiments, for example, were the stoppers deoxygenated to avoid oxygen contamination, how large a headspace was inserted, how long were the samples degassed for (did you have a target oxygen concentration to reach), how was oxygen added back etc

Line 425: Please add in the sampling and storage for the natural abundance nitrate isotope samples.

Line 442: For the nitrite oxidation, I was surprised to see that only a single time point was measured – what influence do the authors think this could have on their rates? Any time lags, flattening off etc would have been missed.

Line 452: In your rate calculations, for the term $n\text{NO}_2^-$, I assume when you say 'measured isotopic composition', you are taking into consideration the likely ^{15}N -nitrate contamination of your ^{15}N -nitrite stock (e.g. Peng et al, 2015, Sun et al, 2017 and Babbin et al, 2020)? As this is a topic that regularly comes up in the literature I suggest you mention that this has been taken into consideration.

Figure 1: Please mention in the caption or mark on the figure itself which are the OMZ and AMZ stations.

Figure 2: As mentioned above a key feature of these experiments is that oxygen was present, thereby allowing you to determine an oxygen consumption rate – this is not the case within the core of the AMZ, so I think it is important to note the starting oxygen concentrations for these experiments (maybe a range in the

caption and in a table in the supplement), this will allow the reader to directly compare these nitrite oxidation rates with those in other papers.

Figure 3: Please clarify in the figure caption the difference between experiment 1, 2, and 3, same experimental setup used? In panels B and C am I correct that the dissolved oxygen concentrations shown on the x axis are those from the starting measurement in the bottle? Please clarify for the reader, the same is true in Figure 4.

Figure 4: This would also be great data to see in a table, how close do your oxygen manipulation experiments get to insitu concentrations?

Table 2: I think it would be beneficial to add insitu oxygen concentrations here

Reviewer #2 (Remarks to the Author):

The paper by Beman et al. is well written and the data are well presented. Oceanic nitrite oxidation is not as well understood as ammonium oxidation and the dissimilatory N-reductions, and a thorough work on the topic is thus essential. As the authors do, it is also very interesting to compare rates, in this case total O₂ uptake and rate of nitrite oxidation. My main worry is that the authors apparently determined O₂ consumption rates during an incubation of 24 h. It has been shown that ocean water respiration rates may increase many-fold when an incubation is extended for more than a few hours (E. Garcia-Robledo et al. 2016, *Front. Mar. Sci.*). These authors also analyzed the effect of O₂ concentration on respiration and found saturation above 0.7 - 1 μM – consistent with the K_m values of low affinity terminal oxidases. In the present paper such a saturation is not found, and there is a large increase with increase in O₂ concentration. Could this also be due to excessively long incubation times?. For any type of time incubation controls of linearity of depletion/accumulation should be performed, but the authors write that they just took starting point and end point after 24 h. My worry is whether the authors are able to supply examples of depletion curves? The rates seem unrealistically high to me – for several of the samples the water would go anoxic within a relatively short time period. For example Station 2 “OMZ edge” have a O₂ depletion rate that would result in anoxic conditions within 2 months. The rates could also be compared with the maximum dissimilatory nitrate reduction rates in the adjacent anoxic water that for similar stations are around 1 nM/L/h, as also found for some of the nitrite oxidation rates in Table 1. O₂ consumption rates have been determined previously at the highly active Station 1 (close to Station M1 of Tiano et al. (Deep-Sea Res. 2014)) and where the present paper finds O₂ consumption rates (V_{max}) of 12, Tiano et al. found a rate of 1 μM/day. Much lower rates were reported by Tiano et al. from a station further offshore.

Specific comments:

Line 38: “Exceptionally” may not be the best word to use here. “very” or “extremely” would be more suitable.

Line 304: The authors claim that they are the first to use bottles with optode dots in OMZ waters, but in the paper by Garcia-Robledo mentioned above they used optode dots with a 1 nM O₂ resolution.

Line 341: The authors suggest that the decrease in OCR at low O₂ concentrations could be due to a shift to denitrification at low O₂ – but the message of the paper is that this nitrate reduction then results in an oxygen demand due to nitrite oxidation in a cryptic nitrite/nitrate cycle..... So net there should be no change!

Line 402. The detection limit of the Firesting is 10 nM – how can you then measure between 0 and 5 nM

Figure 2-5: Find some other colors for the Station 2-3 profiles as they are difficult to distinguish.
The

Table 1. As mentioned in above some more background information for how the data are obtained should be given. Also, I cannot see how a K_s value of 18000 can be calculated with any confidence when only

concentrations up to 18000 were tested. It would be nice to write in the legend that the K_s value for nitrite oxidation is for the O_2 concentration – so that the reader do not need to search the text.

Reviewer #3 (Remarks to the Author):

Beman et al. measure NO_2^- and NH_4^+ oxidation rates in the ETNP OMZ using ^{15}N -tracer incubations. Additionally, they measure OCR in parallel incubations in order to determine the contribution of NO_2^- oxidation to the formation/maintenance of the ETNP OMZ.

I find the entire premise of this manuscript flawed. Beman et al. acknowledge that the water they are incubating is (1) anoxic and (2) that O_2 contamination is unavoidable (see more discussion of (1) and (2) below), however, they then go on to attempt to determine the contribution of *aerobic* NO_2^- oxidation (as the title states, although they are less decisive specifying aerobic vs. anaerobic NO_2^- oxidation in the text) in the *anoxic* zone of the ETNP. By definition, anoxic means that there is no oxygen thus there cannot be any aerobic processes. Beman et al. can reframe their findings as *potential* OCR and aerobic NO_2^- oxidation in the anoxic zone but, given the ubiquity and unavoidability of O_2 contamination in these type of bottle incubations, the OCRs determined in the present study cannot be presented as in situ rates. Beman et al. appear to have not taken all precautions to minimize O_2 contamination during collection and incubation.

(1) Beman et al. state that “accumulations of nitrite in SNMs [are] indicative of anaerobic N cycling” (lines 112-113). I agree that SNMs are indicative of anaerobic N cycling. To my knowledge, Thamdrup et al. (2012) was the first to demonstrate that one of the world’s major OMZs in the eastern tropical South Pacific (ETSP) was not merely low O_2 but “functionally anoxic”. Thamdrup et al. found that O_2 was ubiquitously below the nanomolar detection limit of their O_2 sensor, and, given the range of potential OCRs in the OMZs, the ETSP OMZ must therefore be functionally anoxic. Thamdrup et al. use the language “functionally anoxic” because they acknowledge that there could be still be O_2 present but below detection in the heart of the OMZ meaning any potential *gradient* in O_2 would also be extremely low, with a correspondingly low O_2 flux and OCR, rendering the OMZs “functionally anoxic”. Tiano et al. (2014) went on to confirm this in the ETNP OMZ as well. Consistent with the findings of Thamdrup et al. (2012) and Tiano et al. (2014), when other researchers present NO_2^- oxidation rates from the SNM of an OMZ, they acknowledge that the oxidant is unknown since these regions are anoxic, that is, without oxygen (e.g., Füssel et al., 2012; Peng et al., 2015; Bristow et al., 2016; Babbín et al., 2017; Sun et al., 2017).

(2) Beman et al. state that “typically 1-2 out of 5 replicate AMZ samples” suffered O_2 contamination and that this contamination was virtually unavoidable (lines 300-302). This observation is corroborated by other OMZ researchers and applies to ALL incubation replicates, not just 2 out of 5 replicates. Notably, de Brabandere et al. (2012) demonstrated that even if one was able to avoid O_2 contamination during sampling, incubation vessels themselves will contaminate with O_2 due to O_2 diffusing out of the glass, unless additional precautions are taken. It is also known that the Niskin bottle itself will contaminate with O_2 due to O_2 diffusing out of the plastic of the Niskin bottle. Researchers will often purge water from the anoxic zone with N_2 or He after collection but before incubation in order to remove O_2 contamination from sampling (e.g., Babbín et al., 2017).

Babbín, A. R., Peters, B. D., Mordy, C. W., Widner, B., Casciotti, K. L., Ward, B. B. 2017. Multiple metabolisms constrain the anaerobic nitrite budget in the Eastern Tropical South Pacific. *Global Biogeochemical Cycles* 31(2): 258-271. doi:10.1002/2016gb005407

Bristow, L. A., Dalsgaard, T., Tiano, L., Mills, D. B., Bertagnolli, A. D., Wright, J. J., Hallam, S. J., Ulloa, O., Canfield, D. E., Revsbech, N. P., Thamdrup, B. 2016. Ammonium and nitrite oxidation at nanomolar oxygen concentrations in oxygen minimum zone waters. *Proceedings of the National Academy of Sciences of the United States of America* 113(38): 10601-10606. doi:10.1073/pnas.1600359113

De Brabandere, L., Thamdrup, B., Revsbech, N. P., Foadi, R. 2012. A critical assessment of the occurrence

and extend of oxygen contamination during anaerobic incubations utilizing commercially available vials. *Journal of Microbiological Methods* 88(1): 147-154. doi:10.1016/j.mimet.2011.11.001

Füssel, J., Lam, P., Lavik, G., Jensen, M. M., Holtappels, M., Gunter, M., Kuypers, M. M. M. 2012. Nitrite oxidation in the Namibian oxygen minimum zone. *Isme Journal* 6(6): 1200-1209. doi:10.1038/ismej.2011.178

Peng, X. F., Fuchsman, C. A., Jayakumar, A., Oleynik, S., Martens-Habbena, W., Devol, A. H., Ward, B. B. 2015. Ammonia and nitrite oxidation in the Eastern Tropical North Pacific. *Global Biogeochemical Cycles* 29(12): 2034-2049. doi:10.1002/2015gb005278

Sun, X., Ji, Q. X., Jayakumar, A., Ward, B. B. 2017. Dependence of nitrite oxidation on nitrite and oxygen in low-oxygen seawater. *Geophysical Research Letters* 44(15): 7883-7891. doi:10.1002/2017gl074355

Thamdrup, B., Dalsgaard, T., Revsbech, N. P. 2012. Widespread functional anoxia in the oxygen minimum zone of the Eastern South Pacific. *Deep-Sea Research I* 65: 36-45. doi:10.1016/j.dsr.2012.03.001

Tiano, L., Garcia-Robledo, E., Dalsgaard, T., Devol, A. H., Ward, B. B., Ulloa, O., Canfield, D. E., Revsbech, N. P. 2014. Oxygen distribution and aerobic respiration in the north and south eastern tropical Pacific oxygen minimum zones. *Deep-Sea Research Part I-Oceanographic Research Papers* 94: 173-183. doi:10.1016/j.dsr.2014.10.001

We thank the reviewers for their very helpful comments on our manuscript, which we feel have improved the manuscript considerably. We provide detailed responses to all of their comments below, but wanted to provide an overview of some of the changes that we made.

-First, Reviewers 1 and 2 raised several important points about the OCR measurements. We apologize for not including substantial additional data in our initial submission, as we did measure rates at an earlier time point for the water column profiles, and also have time course data from our experiments. For water column profiles, the majority of OCR values for both sampling time points were highly correlated (with limited exceptions), which is why we didn't include them initially. This information is now provided in the manuscript.

-We also continuously monitored oxygen in multiple incubation bottles during the oxygen manipulation experiments. These data show linear oxygen consumption in the majority of replicates/experiments. (We also note that most of these experiments were ~20 hours—which we should have specified in the original manuscript and now include.) However, nonlinear patterns did emerge in several bottles incubated at <235 nM DO; in these bottles, we actually observed a decline in OCR as DO was consumed during the incubation. This allowed us to use the approach of Tiano et al. (2014) to calculate DO affinities (as highlighted by Reviewer 1), and our results were very similar to theirs (see below for our detailed explanation of this approach, and the results in comparison to earlier work).

-Reviewer 1 also made the excellent suggestion to be more precise about different regions of the water column, which helps clarify several confusing aspects of the manuscript. In particular, this clarifies that: (1) the highest OCR rates were observed above or in the upper portion of the OMZ; (2) many of our samples fall within the SCM (i.e., where oxygen can be supplied via photosynthesis; Garcia-Robledo et al. 2017 in PNAS); and (3) only a limited number of samples were collected below the SCM in the AMZ/ODZ core, where DO is much less likely to be available (but see below). We use this general organization to discuss our results, which helps clarify that the concentrations and rates measured in our experiments are consistent with earlier work.

-As mentioned by Reviewers 1 and 3, we clarify that in the SCM and below, our approach resembles previous work on nitrite oxidation kinetics—which has exposed nitrite oxidizers to oxygen in order to examine their sensitivity (e.g., Bristow et al., Sun et al.). We agree that for our study, like all of these studies, we are measuring the ability of these communities to consume oxygen when it is supplied. However, as we discuss below in response to Reviewer 3, it is important to note that oxygen is regularly supplied to SCM communities via photosynthesis, and that they subsequently consume it. This is an important difference between the SCM vs. the SNM below, and so we discuss results from these depth regions differently.

-Related to this, Reviewer 3 mentions the issue of aerobic vs. anaerobic nitrite oxidation. We clarified this throughout the paper (see detailed locations below), but note that the Reviewer's comments present a one-sided interpretation of the literature that does not fully capture recent research conducted in OMZs. In particular, we explain in more detail the concept of cryptic oxygen cycling (Garcia-Robledo et al. 2017) in the Introduction—i.e., that oxygen is produced via photosynthesis in the SCM in OMZs/AMZs, but it does not accumulate, so it must be rapidly

consumed. We also provide more information from other papers (including those cited by Reviewer 3) raising the idea of oxygen pulses that are drawn down by aerobic processes in OMZs. For example, Tiano et al. (2014) actually show that DO can be present even at 500 m depth in the ETNP—well below the depth of our deepest incubations—further highlighting the fact that DO can be introduced to these regions. The majority of the papers cited by reviewer 3 also explicitly focus on aerobic (not anaerobic) nitrite oxidation (see our detailed responses below).

The key point missed by the Reviewer is that Garcia-Robledo et al., Penn et al., Tiano et al., and others show that oxygen is, in fact, supplied even to anoxic water, but oxygen obviously does not accumulate—so exactly how is anoxia maintained? Throughout the manuscript, we clarify and emphasize that this is the underlying rationale for our work.

We provide more detailed comments below, with reviewer comments shown in italics.

Reviewer #1 (Remarks to the Author):

In the manuscript Constraining the contribution of aerobic nitrite oxidation to oceanic oxygen minimum zone formation, Beman and colleagues present a study that builds on the body of literature looking at the role of nitrite oxidation in low oxygen systems. Specifically, the work demonstrates the potential contribution of nitrite oxidation to oxygen consumption, and how this systemically increased at low (< 2 μM) oxygen levels, and thereby the large role nitrite oxidation may be playing in biogeochemical cycling within ODZs. The overarching hypothesis is noteworthy and certainly of interest to the ODZ community. However in its current format, the experiments and resulting arguments put forward seem a little disconnected from insitu conditions in places, particularly regarding the ODZ core, where oxygen is below detection and a number of other issues deserve attention to help strengthen the study, prior to publication.

We thank the reviewer for recognizing the relevance of our work and for many helpful suggestions. We address the ODZ core samples in our comments directly below.

In order to measure oxygen consumption rates, oxygen needed to be available within your incubations and in all cases shown in Figure 3 it was enough to support all of the nitrite oxidation observed. Is this the case insitu (particularly with regards to the ODZ core)? The starting oxygen concentrations relative to insitu concentrations in these experiments urgently needs to be touched upon and what that means for the nitrite oxidation rates presented. While I understand the value here, and excitement especially with the potential for cryptic oxygen cycling in the SCM, it needs to be highlighted clearly that your incubations do not match insitu conditions in the core and it is more about the potential. Use your incubations to more clearly link to ideas of oxygen availability, and subsequent impacts on biogeochemical cycling.

Some comments directly linked to this:

- Line 300 to 302: here you mention that in one-two of the five replicates in the profile measurements has trace amounts of oxygen, this is not clear to me. What do you mean by trace amounts? Surely you needed oxygen to be measurable to measure your oxygen consumption

rates? I agree that this is unavoidable, but also why it is truly essential that you list the starting oxygen concentrations for these experiments.

- Line 315 to 318: Here you mention the potential for an anaerobic metabolism, but with how the arguments have been built to this point i.e. all nitrite oxidation seen can be aerobic, this seems a little disconnected.

- I would recommend that you try to split up your discussion of experiments that do / dont represent insitu conditions i.e. oxycline and core, I think it will make it much easier for the reader to follow and relate to other literature.

We appreciate these comments a great deal, as they are very helpful in clarifying patterns with depth/oxygen—especially for delineating the importance of different processes at different depths. In particular, we appreciate that the reviewer “understand[s] the value here, and excitement especially with the potential for cryptic oxygen cycling in the SCM.”

We now divide up our presentation by depth. One aspect of this is separating the data from stations 1-3 on Figure 2, and indicating the depth of the OMZ edge and the SCM. As we note above, this more clearly shows that the highest OCR rates are found above or in the upper portion of the OMZ. This also illustrates the fact that many of our samples were collected in the SCM, and just a few (n=5) below the SCM in the AMZ/ODZ core. Throughout the manuscript, we then follow the general pattern of discussing results above the OMZ, on the edge of the OMZ, in the SCM, and below the SCM.

Within the SCM, we think it is important to point out that DO concentrations measured at any given time reflect the balance between DO production via photosynthesis and consumption via respiration and/or nitrite oxidation. In this context, the rates that are found in this region of the water column are probably as relevant or more relevant than DO concentrations (but see our comments directly below). We note in the Introduction (on lines 77-79) that previously measured rates of oxygen production in the SCM can range from 211-2,520 nM per day. We also added an additional one and a half paragraphs within the Results (lines 172-190) describing the OCR rates and comparing them with earlier work. This section is directly relevant to Reviewer 1’s comment below. In particular, while some of our OCR values on the edge of the OMZ are higher than those of Tiano et al., our measurements in the SCM are very much in line with previous studies. Please see our additional comments below.

After we report nitrite oxidation rates, we summarize initial DO concentrations for these incubations (lines 150-154). We apologize for not including this information previously! While this information is presented in Figure 3, we agree that it was definitely an oversight to not summarize this within the text. In brief, our measurements in the SCM and SNM were typically conducted with initial DO values in 100s of nM. For nitrite oxidation, these values are consistent with Bristow et al. but lower than Sun et al. (2019). (This may explain the lack of inhibition observed for our work and Bristow’s in comparison with Sun et al., as inhibition emerges in the μM range, and Sun typically have only a few replicates below 1 μM .) For OCR, our initial DO concentrations are similar to previous work (e.g., Tiano, Kalvelage). This is briefly summarized in the text.

That said, for our study and all previous studies, these values obviously represent potential rates below the SCM, where DO is less likely to be supplied. In the new section on OCR, we now note that our incubations below the SCM reflect the potential of these communities/processes to consume oxygen when it is provided (lines 174-175). As the reviewer mentions, this connects with existing literature both in terms of findings and approach. We apologize for the confusing text regarding “one to two replicates” and modified this text in the Discussion. Here we were specifically referring to samples collected below the SCM (i.e., ODZ core) where we typically saw low but nonzero DO concentrations; however, a few replicates were higher. We revised this text to state that: “In OMZs/AMZs, DO may be photosynthetically produced in the SCM or introduced through physical mixing, but both of these DO sources would be less significant below the SCM. Instead, the introduction of DO during sample manipulation effectively measures the potential for oxygen consumption in samples collected below the SCM. Short of conducting *in situ* incubations, such DO contamination is essentially unavoidable, and is the reason that we also performed oxygen manipulation experiments (Fig. 4)—which also have limitations” (lines 361-367).

We also mention anaerobic metabolism in the Introduction on lines 88-90. We agree that not introducing this earlier in the manuscript makes our later comments seem disconnected. We have additional comments on this issue in response to Reviewer 3 below.

In both the introduction (line 66) and in the discussion (line 157), the authors only refer to / compare to the oxygen consumption rates by Kalvelage et al, 2015, these measurements are largely from shelf ODZs, so are not the best comparison point for your open ocean work. The work however carried out by the group of Niels-Peter Revsbech, and in particular Tiano et al (2014), which carried out oxygen consumption rates in the ETNP (one site very close to one of your own) are a great comparison and raise a number of questions. While there are some similarities there are also some striking differences, particularly with regards to half saturation constants, which Tiano has in the range 10 to 200 nM with the lowest presented here being 1621 nM. The idea of bottle effects presented by Tiano and keeping experiments short for oxygen consumption rates (< 20 hours) to avoid bottle effects, also needs to be commented upon, it is unfortunate that only start and end oxygen concentrations are available from the experiments presented in this paper (any time lags, exponential behavior etc would have been missed) – this points to the potential for your rates to be overestimates, and this needs to be discussed.

These are very useful points and we wanted to take the time to respond in detail, so our comments here are fairly extensive. These comments also address some of the points raised by Reviewer 2 below.

As we note above and discuss in more detail below, we do have time course data from our oxygen manipulation experiments that we can use to examine these patterns, and they are particularly relevant in comparison to Tiano et al.’s data. We agree that Tiano et al.’s findings are a good comparison, and that there are both similarities and differences. However, there are several major methodological differences, too, which is the main reason that led us to not discuss their work extensively in our initial submission. Below we discuss (1) the oxygen levels

used by Tiano, (2) their method for calculating K_m values, (3) use of single K_m values for mixed assemblages, (4) how our K_m values compare with Garcia-Robledo et al. 2016 (answering a point raised by Reviewer 2), and (5) our reasons for calculating the K_m . After this context, we discuss how we use additional data and calculations to address these issues.

(1) First, oxygen levels in Tiano's incubations were very different from in situ values, which is the main reason we did not discuss their work in our initial submission. Based on their Figure 4, Station M1 ODZ core water (300 m) and SCM water (40 m)—which obviously have very low in situ DO concentrations—were incubated in the micromolar range. Tiano et al. also incubated surface (4 m) water at 30-2400 nM—yet in situ DO concentrations at 4 m were 200,000 nM (i.e., 200 micromolar).

Their experiment at 30 m depth had the widest range of DO values (0-1700 nM) and showed sensitivity to oxygen concentrations, with minimum rates at 0-30 nM DO and maximum rates at 1500-1700 nM. However, they did not compute a K_m value for this experiment. In contrast, we had multiple different DO levels for each experiment that spanned the 10s of nM to μ M range, and that we used in our previous calculations (which were similar to the Garcia-Robledo et al. 2016 paper referenced by Reviewer 2 below).

(2) Tiano used a different approach to calculate K_m values, which was to examine changes in OCR as a function of DO over time in incubations. Importantly, they limited this analysis to DO values <500 nM. They calculated parameters for two bottles from 4 m depth at their station M1, and for “both replicate samples the calculation of the K_m value was based on modeling of the 500–0 nmol L⁻¹ O₂ depletion curve.” By definition, calculated K_m values have to be below 500 nM if only considering DO time course data below 500 nM. Yet most of their experiments at M1 show that rates are highest in the micromolar range: rates were maximal at 1500-1700 nM DO at 30 m, 1000-1300 nM at 40 m, and 1400-1450 nM at 300 m.

However, because we did monitor DO in many incubation bottles over time, and because several of these did show decreasing OCR as DO decreased over the experiment, we were able to also use Tiano et al.'s approach to calculate K_m values. For our experiments, K_m values calculated using this approach ranged from 53-127 nM. These values are obviously very similar to Tiano's.

(3) Taking a step back, some of these discrepancies reflect a broader problem that we previously mentioned in our manuscript, but should have discussed with greater care: assuming a single K_m for the consumption of oxygen is a substantial oversimplification. In fact, Tiano et al. clearly state that “A simple Michaelis–Menten (or Jassby and Platt) model is thus an oversimplification.” This applies to organisms respiring organic carbon, but this also applies to nitrite oxidation. As Bristow et al. (2016) show, even within nitrite oxidizers alone, the “oxygen response likely represents the response of a mixed community potentially carrying a variety of terminal oxidases with different K_m values.” We now include more information on different approaches to calculation of K_m values in Supplementary Note 1, and mention some of the underlying complexity (assemblages, enzymes, substrates) that results in differences in K_m values calculated using different approaches.

(4) We previously used an approach similar to Garcia-Robledo et al. (2016), who conducted incubations at multiple DO levels (albeit on different days). Our observations resemble those of Garcia-Robledo et al., who, as Reviewer 2 states below, report saturation in the μM range. However, Garcia-Robledo et al. only had one incubation for which they were able to use Tiano et al.'s approach to calculate a low-level K_m . Another important point is that their r^2 value for their experiments is lower due to scatter in their rate measurements: in particular, they observed a linear increase in rates with increasing DO that actually extended up to $\sim 2 \mu\text{M}$, but then rates were lower at $\sim 5 \mu\text{M}$ DO and higher. Finally, it is worth noting that Kalvelage et al. found DO sensitivity at higher DO concentrations in the 10s of μM . This was a consistent experimental observation by an excellent group of researchers, which suggests that DO sensitivity for respiration is quite complex.

There were several related issues highlighted by the reviewers that we address in our revised manuscript. First, most of our incubations were around 20 hours in length (as we noted above, and now specify in Table 1 in the manuscript)—with the exception of Station 1. The Station 1 OMZ edge experiment extended for over 30 hours due to the fact that we could not access the cold van during the hurricane. As the reviewers keenly observed, the kinetic parameters for this experiment are unreasonably high (probably due to length, and possibly due to disturbance during the hurricane). We therefore removed this experiment from the manuscript. Second, there was a typo for one of our K_m values that Reviewer 2 highlighted below. We apologize for this very stupid mistake, but the K_m value for the Station 3 SCM experiment was 1800 nM, not 18,000. Finally, we found that another solution for the Station 2 SCM experiment produced an equally good fit. As a result, four of our experiments have overall K_m values around 1-2 μM (and we also report the high affinities calculated using Tiano et al.'s approach), with two experiments slightly higher than this, and one experiment higher still. We now include a Supplementary Note 2 that compares these results with earlier work. We also include information on the kinetic parameters for nitrite oxidation in Supplementary Note 2, as several reviewers noted that comparison with earlier work would be useful. In general, our overall K_m values for OCR are in line with, but slightly higher than, Garcia-Robledo and Tiano et al.'s observations at different DO levels. However, they are lower than what was observed by Kalvelage et al. And again, our new low-level affinity values are very much in line with Tiano et al. because we used their approach where applicable. Our values for nitrite oxidation are also consistent with the literature.

(5) After these long explanations, we note that the main goal of these calculations was simply to use the low end of the curves to calculate how the contribution of nitrite oxidation increases with decreasing DO. Our goal was not to argue for a single K_m for the whole community, and we should have handled the presentation of our calculations much more carefully. We had actually previously made multiple other calculations—which we now include—but felt that the kinetic approach was straightforward (obviously it isn't).

More importantly, the data themselves are critically important in constraining the contribution of nitrite oxidation to oxygen consumption, while the calculations compliment our overall point.

(This is why we placed some of the information above in the Supplement.) Our experiments place an upper limit on the DO level at which nitrite oxidation can account for all DO consumption (ca. 393 nM), but the data themselves point to the fact that this more typically occurs at <100 nM DO. Throughout the manuscript, we follow the recommendation of Reviewer 1 below to state specific DO values, and we emphasize the data versus the calculations. Finally, it is important to note the consistency between results from profiles and experiments, and the fact that the ratio of nitrite oxidation to OCR approaches 100% for both. This means that our OCR measurements cannot be substantially overestimated in either set of experiments, and that the overall pattern is robust.

To further emphasize our points in the paper, we included the following additional information and calculations:

- As noted above, we used the approach of Tiano to calculate DO affinities at low DO concentrations based on the relevant DO time courses.
 - We applied regression to the relationships in Figure 3 to estimate when nitrite oxidation can account for all OCR (based both on profiles and experiments).
 - We also included results for regression (based on $1/[DO]$) for individual experiments.
- Again, we emphasize that these are all broadly consistent because they are based on our data, and our data provide important constraints.

Specific comments

Title: I find the use of the word ‘formation’ quite strong, yes your experiments indicate that nitrite oxidizers have a potentially important role to play in oxygen consumption, but without sampling on a larger spatial and temporal scale, along with modelling, the word formation seems a bit too large of a step. Throughout, regarding the link to formation and maintenance of the ODZ, this needs to be toned down in places.

We agree with this comment when thinking about the larger scale oceanographic processes that are important in OMZ/ODZ formation. We changed the title to remove ‘formation,’ and instead focus on the role of nitrite oxidation in consuming oxygen in OMZs. This also clarifies the broader context of our work—i.e., that oxygen is supplied biologically and physically to OMZs and must be removed somehow. Throughout the manuscript, we typically changed ‘formation’ to refer to the maintenance of low DO concentrations in OMZs.

Line 51, 63, 80: throughout, language such as low, intrinsically low, trace is used (these are just a few examples), please try to be quantitative wherever possible, these words have very different meanings for different individuals.

This is a good point and we have clarified this throughout by using specific numbers. For example, we added DO concentration values to the abstract. In the Intro, the concentrations typically used to define OMZs are specified after the more general starting sentences. We agree that this is an issue that arises repeatedly in the literature—for example, some of the studies referenced by the reviewers refer to a variety of oxygen levels in the micromolar range as ‘low.’

Our key findings are based on work in the 10-100s of nM range, not the micromolar range.

Line 62 to 64: references are needed at the end of this sentence.

We added a sentence prior to this to note that oxygen can be produced or introduced into OMZs (lines 62-64). We included relevant references in this new sentence, and also added relevant references to the end of the sentence identified by the reviewer (now lines 64-66).

Line 85: 'declining rates with declining DO' here and in this paragraph in general it is surprising that you don't take the time to explain in more detail the kinetics work done to date (e.g. Sun et al, 2017, Bristow et al, 2016) introducing the half saturation constants, high oxygen affinity etc, as it is this work you are attempting to build on.

This is a good suggestion, and we added a sentence to this paragraph (lines 86-88) noting the high oxygen affinities measured in earlier work. As noted above, we include more detailed information and discussion in the Supplement, because there are some differences in approach and results among studies that are important to discuss.

Line 117: it would be beneficial for the reader to add in the depth of the oxic-anoxic interface (for all stations)

Following the recommendation of the reviewer above, we split out the different stations into a series of plots that help clarify the key depths. These key depths are now specified on Figure 2.

Line 134: My understanding is that your definition of an OMZ station is that oxygen drops below 20 μ M, looking at Figure S1 this doesn't seem to be the case for Stations 4 and 6? Is this just due to the depth ranges plotted?

Yes, this is due to the depth range plotted. We now mention this in the figure caption.

Line 275: this is the first mention of an anaerobic metabolism, this needs some context prior to this. Is there any other evidence in the metagenomes for an anaerobic metabolism? It would be great to hear more about your findings from these metagenomes, this data seems a little underutilized.

As we mentioned above, we added background on anaerobic metabolism to the Introduction in order to provide more context. Obviously the metagenomes are a rich dataset, and here we are using them to support points made in the paper, so we kept our analysis fairly focused (and are conducting many additional analyses). However, following this suggestion, we did include the relative abundance of formate dehydrogenase and nitrate reductase genes from *Nitrospina* in an online Table (and moved the chlorite dismutase data there). We added several sentences to the Results briefly describing these results (lines 333-338) and they are included in a Supplementary

Table.

Line 318 to 319: how do the rates of oxygen production observed in Garcia-Robledo et al, 2017 align with what you have seen / expect?

One notable finding from Garcia-Robledo 2017 is that the oxygen production rates can be quite significant in the SCM. As mentioned above, we added these values to the Introduction. When we present our OCR results, we now note that our consumption rates fall within this range.

Line 342: I suggest you link to the work of Sun et al, 2017 here and the Km for nitrite.

We thank the reviewer for this suggestion, and added a sentence just after this noting that “The fact that nitrite oxidation rates increase with increasing nitrite into the micromolar range... provides additional support for this idea” (lines 414-416).

Line 397: which experiments are you referring to when you say ‘deoxygenation experiments’?

We corrected this in the text to ‘oxygen manipulation experiments.’

Line 402: please clarify what you mean by ‘within 5nM of 0’ – is the 0 you refer to here 10nM (the detection limit mentioned)?

Another reviewer commented on this, and we apologize for the confusion. We did observe values <10 nM in some DO time courses, but we have changed this to 10 nM, since that is the manufacturer determined limit.

Line 410 to 412: the initial DO concentrations in the BOD bottles, how close were these to insitu values (the detection limit of your SBE-43 will also be important here, line 374)? Were the BOD bottles cleaned prior to use to avoid contamination with organics which could lead to bottle effects (e.g. Tiano et al, 2014)

As mentioned above, we now report the starting DO values in the Results, particularly for measurements in the SCM and below. Starting values in the SCM and below were typically in the 100s of nM, in line with previous work. The bottles were cleaned between use; we added this info to the Methods and apologize for the oversight.

Line 415 to 423: a lot more detail is needed here to allow someone to repeat these oxygen manipulation experiments, for example, were the stoppers deoxygenated to avoid oxygen contamination, how large a headspace was inserted, how long were the samples degassed for (did you have a target oxygen concentration to reach), how was oxygen added back etc

We apologize for the lack of information here, although we note that some of these specific details are not always included in other papers. We added additional information to the Methods (including everything listed above) and are happy to supply any additional information that would be useful.

Line 425: Please add in the sampling and storage for the natural abundance nitrate isotope samples.

We added this information to the Methods (filtered and frozen).

Line 442: For the nitrite oxidation, I was surprised to see that only a single time point was measured – what influence do the authors think this could have on their rates? Any time lags, flattening off etc would have been missed.

We have conducted a fair amount of these measurements and definitely agree that there may be interesting behavior over time. For example, the DO depletion curve approach that can be used for OCR would also be relevant for calculating low K_m values for nitrite oxidation. However, our main focus was on the comparison with OCR as a function of oxygen, and we elected to spend more of our replicates and sampling on different oxygen levels. It would be difficult to sample the same bottles at multiple time points without oxygen contamination, which then requires additional replicate bottles for sampling over time.

Line 452: In your rate calculations, for the term $n\text{NO}_2^-$, I assume when you say ‘measured isotopic composition’, you are taking into consideration the likely ^{15}N -nitrate contamination of your ^{15}N -nitrite stock (e.g. Peng et al, 2015, Sun et al, 2017 and Babbin et al, 2020)? As this is a topic that regularly comes up in the literature I suggest you mention that this has been taken into consideration.

This is a good point and an issue that we have encountered before (and may have been the first to identify for nitrite oxidation rate measurements; Beman et al 2013 ISMEJ). We now mention that this was incorporated into the calculation.

Figure 1: Please mention in the caption or mark on the figure itself which are the OMZ and AMZ stations.

Thanks for the suggestion; we modified the symbols on the figure and the figure caption to denote the AMZ versus OMZ stations.

Figure 2: As mentioned above a key feature of these experiments is that oxygen was present, thereby allowing you to determine an oxygen consumption rate – this is not the case within the

core of the AMZ, so I think it is important to note the starting oxygen concentrations for these experiments (maybe a range in the caption and in a table in the supplement), this will allow the reader to directly compare these nitrite oxidation rates with those in other papers.

As mentioned above, we included text in the paper that notes the starting values for these measurements, and how they compare with the literature.

Figure 3: Please clarify in the figure caption the difference between experiment 1, 2, and 3, same experimental setup used? In panels B and C am I correct that the dissolved oxygen concentrations shown on the x axis are those from the starting measurement in the bottle? Please clarify for the reader, the same is true in Figure 4.

We clarified in the figure caption that some Stations had multiple experiments, and that these were conducted at different depths. We clarify that these are the average oxygen concentrations, which was used by Garcia-Robledo, Tiano, and others (and unclear in other studies).

Figure 4: This would also be great data to see in a table, how close do your oxygen manipulation experiments get to insitu concentrations?

We included the in situ oxygen concentrations in Table 1 and discuss the relative concentrations used in the experiments within the text (they are also shown on the figures). Again, incubations in the μM range were included in all experiments for purposes of comparison with earlier work and across experiments, but most of our measurements were in the 100s of nM DO, with some dropping into the 10s of nM.

Table 2: I think it would be beneficial to add insitu oxygen concentrations here

We also added these in situ oxygen concentrations to the Table.

Reviewer #2 (Remarks to the Author):

The paper by Beman et al. is well written and the data are well presented. Oceanic nitrite oxidation is not as well understood as ammonium oxidation and the dissimilatory N-reductions, and a thorough work on the topic is thus essential. As the authors do, it is also very interesting to compare rates, in this case total O₂ uptake and rate of nitrite oxidation.

We thank the reviewer for these positive comments.

My main worry is that the authors apparently determined O₂ consumption rates during an incubation of 24 h. It has been shown that ocean water respiration rates may increase many-fold when an incubation is extended for more than a few hours (E. Garcia-Robledo et al. 2016, Front. Mar. Sci.). These authors also analyzed the effect of O₂ concentration on respiration and found saturation above 0.7 - 1 μ M – consistent with the K_m values of low affinity terminal oxidases. In the present paper such a saturation is not found, and there is a large increase with increase in O₂ concentration.

This is a very interesting and relevant study that we neglected to cite in our original submission, and we've added it to the Discussion and references. This is also very good point, and please see our long responses to reviewer 1 above, which cover some of these same topics in detail (in particular point #4 in our long response).

We did include an earlier sampling time point for the water column profiles, and OCR values were consistent throughout these incubations, with a few exceptions (see text on lines 482-487 in the methods)—which is why we didn't include this previously. Most of our experiments were also 20 hours or less, with the key exception of the Station 1 experiment—which was unavoidably long and we removed from the manuscript.

As noted above in the response to Reviewer 1, we also have time course data from the experiments, most of which show linearity. The key exceptions were all at DO concentrations <235 nM and show that OCR rates decline over time and with decreasing DO. These data are useful for calculating high DO affinities—which are in line with Tiano et al., and the K_m values are lower than Garcia-Robledo et al. 2016 due to the approach used. More detail is included in our response above to Reviewer 1, and we also discuss this the Supplement. As we note above, the saturation in OCR for both Tiano et al. and Garcia-Robledo et al. 2016 also occurs in the range we observe (closer to 2 μ M than 1 μ M based on the actual data reported in both papers), and lower than Kalvelage. We also note this in the Supplement.

Could this also be due to excessively long incubation times?. For any type of time incubation controls of linearity of depletion/accumulation should be performed, but the authors write that they just took starting point and end point after 24 h. My worry is whether the authors are able to supply examples of depletion curves? The rates seem unrealistically high to me – for several of the samples the

water would go anoxic within a relatively short time period. For example Station 2 “OMZ edge” have a O₂ depletion rate that would result in anoxic conditions within 2 months.

We apologize for not including these data initially, but as mentioned above, we did continuously monitor DO in multiple incubation bottles throughout all the experiments. We tested all for linearity and found that most DO time courses were linear. Those that were nonlinear did not accelerate, they showed a decrease in OCR as DO decreased throughout the incubations. This pattern allowed us to calculate affinities following Tiano et al.

Re the specific example of station 2, if the reviewer is referring to profiles, the separation of figure panels may help clarify that the rate is quite reasonable (the highest rate is well above the OMZ). We emphasize this in the paragraph added to the results where compare our rates to those in the literature (lines 172-190). If the reviewer is referring to the experiments, the overall v_{\max} indicates that this would take closer to 5 months (and is probably enhanced by bubbling). However, it absolutely critical to note that oxygen could be readily supplied to this depth (90 m) via photosynthesis and especially mixing. 90 m sits at the bottom of the primary chl max, well above the SCM. Mixing may easily reach 90 m—e.g., there was a hurricane in this region at the time of our sampling. We want to underline the point that OMZs are dynamic, and different depth horizons within OMZs can be very different. The core is obviously much more isolated than the edge—and the edge can be found at fairly shallow depths. We hope that this overall idea is now presented more clearly in the manuscript.

The rates could also be compared with the maximum dissimilatory nitrate reduction rates in the adjacent anoxic water that for similar stations are around 1 nM/L/h, as also found for some of the nitrite oxidation rates in Table 1. O₂ consumption rates have been determined previously at the highly active Station 1 (close to Station M1 of Tiano et al. (Deep-Sea Res. 2014)) and where the present paper finds O₂ consumption rates (V_{\max}) of 12, Tiano et al. found a rate of 1 μ M/day. Much lower rates were reported by Tiano et al. from a station further offshore.

Please see our long comments above in response to Reviewer 1 that are focused on the Tiano et al. paper. When comparing our water column profile—not the experiment—to Tiano et al., the results are generally quite similar. We make these comparisons in the revised manuscript. Importantly, only a few measurements in the upper water column are higher than Tiano et al.’s values. In addition, we note again the range of oxygen values they used in comparison to reality, which may skew their values. Finally, as noted above, we removed the Station 1 experiment owing to the unavoidable length of it.

We also added a brief comparison to nitrate reduction rates (lines 414-416), as these are a relevant and useful comparison. We thank the reviewer for suggesting this.

Specific comments:

Line 38: “Exceptionally” may not be the best word to use here. “very” or “extremely” would be

more suitable.

Following the recommendation of Reviewer 1, we changed this to the actual DO concentration at which we measured nitrite oxidation to be dominant in the Station 2 SCM experiment.

Line 304: The authors claim that they are the first to use bottles with optode dots in OMZ waters, but in the paper by Garcia-Robledo mentioned above they used optode dots with a 1 nM O₂ resolution.

Here we referring to the fact that we used optode dots along with nitrite oxidation measurements, not the first to use dots overall, so we agree that this wasn't clear! We removed the first part of this sentence to simply note the inclusion of optode dots (now on lines 368-370).

Line 341: The authors suggest that the decrease in OCR at low O₂ concentrations could be due to a shift to denitrification at low O₂ – but the message of the paper is that this nitrate reduction then results in an oxygen demand due to nitrite oxidation in a cryptic nitrite/nitrate cycle..... So net there should be no change!

We revised this text to make this clearer. We agree that our previous use of the word 'jumpstart' implies that the nitrite oxidation rates would increase, leading to no net change. Now we simply state that nitrate reduction would provide a needed substrate for nitrite oxidation.

Line 402. The detection limit of the Firesting is 10 nM – how can you then measure between 0 and 5 nM

Please see our response to Reviewer 1 above; we changed this 10 nM.

Figure 2-5: Find some other colors for the Station 2-3 profiles as they are difficult to distinguish.

We appreciate this recommendation, and throughout all the Figures we changed the Station 3 color to a darker shade of blue that is easier to distinguish.

Table 1. As mentioned in above some more background information for how the data are obtained should be given. Also, I cannot see how a K_s value of 18000 can be calculated with any confidence when only concentrations up to 18000 were tested. It would be nice to write in the legend that the K_s value for nitrite oxidation is for the O₂ concentration – so that the reader do not need to search the text.

As we noted above, this was a very stupid mistake: 18000 nM was a typo—it is 1800—and we fixed this. We also noted in the caption that the K_m is for oxygen.

Reviewer #3 (Remarks to the Author):

Beman et al. measure NO₂⁻ and NH₄⁺ oxidation rates in the ETNP OMZ using ¹⁵N-tracer incubations. Additionally, they measure OCR in parallel incubations in order to determine the contribution of NO₂⁻ oxidation to the formation/maintenance of the ETNP OMZ.

*I find the entire premise of this manuscript flawed. Beman et al. acknowledge that the water they are incubating is (1) anoxic and (2) that O₂ contamination is unavoidable (see more discussion of (1) and (2) below), however, they then go on to attempt to determine the contribution of *aerobic* NO₂⁻ oxidation (as the title states, although they are less decisive specifying aerobic vs. anaerobic NO₂⁻ oxidation in the text) in the *anoxic* zone of the ETNP. By definition, anoxic means that there is no oxygen thus there cannot be any aerobic processes.*

We agree with some aspects of this, but these comments do not accurately represent the full extent of what occurs biogeochemically in OMZs/AMZs based on the literature. In particular, these comments confuse concentrations and rates: while it is true that no oxygen is typically *measured* in AMZs (although sometimes it can be; see below), that is because oxygen that is *produced/introduced* is rapidly *consumed*. This is the important and interesting issue raised by earlier work that we are focused on here. In short, yes, these regions are functionally anoxic, but how/why are they functionally anoxic, and what are the implications?

We modified the Introduction to make this as clear as possible earlier in the manuscript. Papers by Garcia-Robledo et al. 2017 in PNAS and Penn et al. 2019 in PNAS are critically important in establishing the idea that because DO is produced in and/or introduced to OMZs, it must be consumed by aerobic processes in order to maintain functional anoxia. Please see the new text added to the second and third paragraphs in the Introduction (lines 62-81) about DO production/introduction to AMZs, and see these papers for additional information and context.

*Beman et al. can reframe their findings as *potential* OCR and aerobic NO₂⁻ oxidation in the anoxic zone but, given the ubiquity and unavoidability of O₂ contamination in these type of bottle incubations, the OCRs determined in the present study cannot be presented as in situ rates. Beman et al. appear to have not taken all precautions to minimize O₂ contamination during collection and incubation.*

See our comments above in response to Reviewer 1 about thinking of the SNM/AMZ measurements as potentials. We agree with this and now note in both the Results and the Discussion that rates below the SCM should be considered potentials (lines 174-175 and lines 361-367). However, this potential is important to understand and quantify. Please see below regarding experimental approaches.

(1) Beman et al. state that “accumulations of nitrite in SNMs [are] indicative of anaerobic N cycling” (lines 112-113). I agree that SNMs are indicative of anaerobic N cycling. To my knowledge, Thamdrup et al. (2012) was the first to demonstrate that one of the world’s major OMZs in the eastern tropical South Pacific (ETSP) was not merely low O₂ but “functionally anoxic”. Thamdrup et al. found that O₂ was ubiquitously below the nanomolar detection limit of

*their O₂ sensor, and, given the range of potential OCRs in the OMZs, the ETSP OMZ must therefore be functionally anoxic. Thamdrup et al. use the language “functionally anoxic” because they acknowledge that there could be still be O₂ present but below detection in the heart of the OMZ meaning any potential *gradient* in O₂ would also be extremely low, with a correspondingly low O₂ flux and OCR, rendering the OMZs “functionally anoxic”. Tiano et al. (2014) went on to confirm this in the ETNP OMZ as well.*

Consistent with the findings of Thamdrup et al. (2012) and Tiano et al. (2014), when other researchers present NO₂- oxidation rates from the SNM of an OMZ, they acknowledge that the oxidant is unknown since these regions are anoxic, that is, without oxygen (e.g., Füssel et al., 2012; Peng et al., 2015; Bristow et al., 2016; Babbin et al., 2017; Sun et al., 2017).

We are familiar with this work, much of which is cited and discussed in the paper. However, the interpretation of these papers by the reviewer isn't accurate. The entire premise of the Bristow et al. study is, in fact, to examine the oxygen affinity of ammonia and nitrite oxidation (as mentioned above by other reviewers). Füssel et al. and Sun et al. also specifically include data and figures displaying relationships between oxygen concentrations and nitrite oxidation, noting that nitrite oxidation occurs at low but non-zero oxygen concentrations. Füssel only raise anaerobic nitrite oxidation as a possibility in the second-to-last sentence of their paper. Sun et al. 2017 specifically focus on the dependence of nitrite oxidation on nitrite and oxygen (e.g., see the title of their paper), including just a few sentences on alternative oxidants.

Sun et al. 2017 do cite Babbin et al. 2017, who performed some interesting experiments adding iodate as a potential alternative oxidant. However, Bess Ward's research group has backed off on this idea considerably; see recent papers by Sun et al. that we cite and which note that iodate concentrations are very low in OMZs.

Please also see our comments above in response to Reviewer 1 re the Tiano et al. study, and specifically the depths they sampled and the oxygen concentrations at which different samples were incubated. As we note above and in our revised manuscript, it is also critically important to recognize that Tiano et al. did observe DO at 500 m depth in the ETNP. This is in the core of the AMZ/ODZ, and well below our measurements. Their findings clearly highlight the fact that DO is produced/introduced to AMZs, and so must be consumed by aerobic processes.

(2) Beman et al. state that “typically 1-2 out of 5 replicate AMZ samples” suffered O₂ contamination and that this contamination was virtually unavoidable (lines 300-302). This observation is corroborated by other OMZ researchers and applies to ALL incubation replicates, not just 2 out of 5 replicates. Notably, de Brabandere et al. (2012) demonstrated that even if one was able to avoid O₂ contamination during sampling, incubation vessels themselves will contaminate with O₂ due to O₂ diffusing out of the glass, unless additional precautions are taken. It is also known that the Niskin bottle itself will contaminate with O₂ due to O₂ diffusing out of the plastic of the Niskin bottle. Researchers will often purge water from the anoxic zone with N₂ or He after collection but before incubation in order to remove O₂ contamination from sampling (e.g., Babbin et al., 2017).

We in fact conducted oxygen manipulation experiments that purged water with He, exactly as the reviewer mentions. We are aware of the issues of sampling, which is the reason that we also conducted these experiments. However, most earlier work was then done blindly without knowing the actual DO concentrations in incubations, which is why oxygen contamination is a key issue—especially for anaerobic processes like anammox and denitrification. However, because we have sensor spots in the bottles, we are, in fact, able to detect any oxygen contamination over time. Please see our response to Reviewer 1 about this particular statement in the paper, which we agree was unclear and have rephrased. We also included additional methodological information following the comments of Reviewer 1.

We generated a range of oxygen concentrations to examine rates as a function of oxygen, as this is the intent of our study. Note that we were not focused on complete anoxia: while this is an interesting open question, we are focused on how produced/introduced DO is consumed in OMZs/AMZs (which is hopefully clear at this point).

We also note that while bubbling helps resolve issues with oxygen, it generates its own set of issues (e.g., those raised by other reviewers above). We therefore used both bubbled and non-bubbled incubations. Most studies have not used this dual approach, and our results from each approach agree. Please see lines 361-373 of the revised manuscript.

REVIEWER COMMENTS

Reviewer #1 (Remarks to the Author):

Firstly, I would like to thank the authors for the time and consideration they put into the revision and reviewer response, which I believe has resulted in an improved manuscript. However, while the overall message is much clearer and aligned with the environment sampled, clarity is still needed in some places, which I outline below.

Line 38-39: 'Nitrite oxidation was dominant under DO concentrations < 393 nM', this sentence needs to be clarified, what do you mean by dominant (I assume you mean could consume all available DO, but that is not clear as currently written), where in the water column is this relevant (SCM and SNM), how was this determined.

Line 79, 99, 123, 423: these are just a few examples, the authors mention nitrate reduction a lot, as they should, but I find it surprising that they never touch on the oxygen sensitivity of this process (e.g. Kalvelage et al, 2011; Bristow et al, 2016 etc) and the potential overlap with nitrite oxidation making this even more of an exciting topic.

Line 88 to 90: It is not clear to me what omic data the authors are referring to in the literature cited as evidence for alternative electron donors and acceptors for NOB? What about data beyond omics that suggests a role for anaerobic nitrite oxidation, a more thorough introduction seems warranted here.

Line 92 to 94: while I 100% agree that variations in genomic content is a potential explanation here, high in situ rates under low DO (putting a number here would be helpful for the reader) could also surely be explained at least to some extent by oxygen contamination – which surely only strengthens your study as you measured oxygen in your incubations. I know that you come to this point in your discussion but it also seems relevant here.

Results / Discussion: as currently presented there is a lot of discussion in the results section (e.g. 178 to 190, 230 to 238 etc) and the discussion more of a summary in places, maybe this could become a single results / discussion section? (some restructuring would be needed to make this work)

Line 151-152: I value that the authors have included the starting DO concentrations in their incubations, and then later for OCR (line 175) state that these rates are likely potential due to this, why is there not a similar statement for nitrite oxidation rates? Here seems like a good place to do it, to be upfront.

Line 188: if you want to include the value from Namibia here, I think you need to be clear that it is from shelf waters and not offshore deep waters like your own work and that for the values you present from Kalvelage et al, 2015.

Line 220 and 221: what is the basis for this being the cutoff for potential rates?

Line 304: why are you only correlating your DNA / RNA with the natural abundance data, surely your rates are more appropriate with respect to timescale.

Line 308 to 342: this is super interesting! Why is a plot of the individual ASVs not included in the manuscript? This really helps bring your ideas together. It would also be valuable here to contrast your metagenomic findings to that of other OMZ studies such as Sun et al, 2019 in more detail (e.g. terminal oxidases, nitrate reductase etc), apart from one sentence regarding chlorite dismutase, this is very much a results section and it would be valuable for the reader for you to compare / contrast your findings to work to date.

Line 384: an additional reference here would be Larsen et al, 2016 In situ quantification of ultra-low O₂ concentrations in oxygen minimum zones: application of novel optodes. L&O Methods 14 (784-800) (see

Figure 6 and text)

Line 396: please clarify what you mean by dominant here.

Line 440: what experiments were conducted in which year?

Line 463 to 464: is the 10nM detection limit taken from the Lehner et al paper? As I understand it that paper is discussing completely different optode chemistries to that of the commercially available trace optodes available from Pyroscience. What spots / chemistry did you use for the oxygen manipulation experiments?

Line 471: this sentence is not at all clear to me 'within 10nM of zero'?

Line 473 to 475: this seems out of place and no justification is provided

Figure 3: in the caption I think you need to link symbol to location in the water column for panel C e.g. Station 2 (SCM (closed purple) and OMZ edge (open purple)

Figure 4: would it be possible to reorganize the panels so that OMZ edge, SCM and SNM aligned vertically, this would make it easier to compare.

Table 1: it would be valuable here to include the errors on your K_m / V_{max} numbers

Table 2: DO, I assume this is from the SBE 43 on the CTD? If yes, what is the detection limit of the sensor, this is needed so that readers do not assume that oxygen was measurable at all depths sampled e.g. $0.3 \mu\text{M}$ at the SNM. More information is needed here regarding the functional genes e.g. nitrite oxidoreductase (nirB?). All of these comments are also relevant for Table S1.

Reviewer #2 (Remarks to the Author):

Beman et al., Constraining the contribution....

The authors have answered very appropriately to the questions asked from the three reviewers of the first version. I thus feel that the manuscript now is in a good shape. However, I still recommend a few minor changes.

Intro: A small typo: refs 10 and 11 should be superscript.

Figure 2: It should be mentioned in the figure legend (and also more specifically in the main text) that the OCR data are potential rates. Most would feel it strange that the authors find positive OCR values in the anoxic layers!

Line 181: I have had a look on the photosynthesis data of ref. 15, and they fall far below the values given here. At an in situ light level the photosynthesis rate at noon is below 5 and 10 $\text{nmol L}^{-1} \text{h}^{-1}$ in the ETNP and ETSP, respectively, so rates are below 50 and 100 per day, respectively. Respiration rates are variable but more realistic. So it would be better not to write that photosynthesis and respiration rates are in the same range.

Line 215-217: It is actually not very clear that nitrite reduction is decreased at low O_2 in some of the experiments, and I thus do not think that the authors should use the word "steep"

Line 250 - 264: This is too speculative, and I suggest the section modified. It is clear that nitrite oxidation will never account for all oxygen consumption, and any extrapolation of consumption rates down in O_2 concentration to obtain 100% consumption by nitrite oxidation has no meaning! But that nitrite oxidation becomes progressively more important by decreasing O_2 is correct!

Lines 270-271: "NOB are therefore effective at scavenging DO (Figs. 3 and 4; ref. 15), and the concentration range over which this occurs in our experiments matches model predictions ($0.2\text{-}1 \mu\text{M DO}$)" – I do not like this sentence, as DO scavenging occurs throughout all O_2 concentrations, but at varying importance – so rewrite.

Line 403-5: Nitrite oxidation is also a respiration, so please write heterotrophic respiration in contrast to OCR by nitrite oxidation.,

Reviewer #3 (Remarks to the Author):

Beman et al. have addressed most of the comments of the reviewers. There are still a few things that could be improved.

Reviewers 1 and 2 both wanted to know if there were lags or non-linear behavior during the course of the incubations. The authors added discussion of the trends in the text but it is still unclear. This point about linearity is important to make clear so the authors should present in figures at least some of the timecourses that they have measured – moles of NO₂⁻ oxidized vs. time – in the supplemental would be fine.

Also related to this discussion of linearity in the timecourses:

Lines 482-485 "slopes ranging from 1.06-1.19" What are the units for these values? These slopes are not a large enough range to represent the range in rates presented in Fig. 2.

Lines 485-487 "The only exceptions were three sampling depths...showing nonlinearity, and so we report data from measurements conducted at similar depths...the previous day." Did all the rest of the "profile" rate measurements occur on a single day? Why was this so important? Does this mean the only nonlinear timecourses observed were from the second day (I say "second" only because the authors refer to the "previous" day, not because I understand what the authors mean here)? And these nonlinear timecourses were then...disregarded?

Reviewer 1 suggested adding the in situ DO concentrations, specifically in the context of the detection limit of the Seabird SBE-43. The authors added some of this information in Table 1 however it could be further clarified:

- 1) It should be labeled "in situ DO measured by SBE-43" in Table 1 otherwise it is unclear if "starting DO" is the DO measured in the incubation bottle after collection.
- 2) The authors should include the in situ DO concentrations for the "profile" incubations as well.
- 3) The SBE-43 has a published accuracy of +/-2% of saturation. The authors are reporting DO concentrations well below this. If they did additional calibrations to resolve lower DO concentrations with the SBE-43 they should explain what they did. Otherwise, "below detection" should be stated.

Also related to detection limits:

Lines 471-471 "our lowest values were consistently within 10nM of 0" This suggests that the values <10nM are significant. Instead the authors should state "our lowest values were consistently below the detection limit of the FireSting"

We thank the reviewers for their very helpful comments on our revised manuscript, which we feel have further improved the manuscript. Below we provide detailed responses to all of their comments, with our responses in blue text.

Reviewer #1 (Remarks to the Author):

Firstly, I would like to thank the authors for the time and consideration they put into the revision and reviewer response, which I believe has resulted in an improved manuscript. However, while the overall message is much clearer and aligned with the environment sampled, clarity is still needed in some places, which I outline below.

We thank the reviewer for all the helpful comments, and respond to everything below.

Line 38-39: 'Nitrite oxidation was dominant under DO concentrations < 393 nM', this sentence needs to be clarified, what do you mean by dominant (I assume you mean could consume all available DO, but that is not clear as currently written), where in the water column is this relevant (SCM and SNM), how was this determined.

We apologize for the lack of specificity here and thank the reviewer for noting this. We are definitely constrained by word limits in the Abstract, but have addressed this in our revision (which puts us slightly over the word limit). In thinking about this and some of the other reviewer comments, 'dominant' would more accurately refer to >50% of overall OCR, at which point nitrite oxidation consumes more DO than any other process. We decided not to use 'dominant' in the Abstract, and instead refer to consumption of all DO. We also note that this occurs in and below the secondary chlorophyll maximum. As to how this was determined, the sentence before this one mentions our approach (i.e., 'parallel measurements').

Line 79, 99, 123, 423: these are just a few examples, the authors mention nitrate reduction a lot, as they should, but I find it surprising that they never touch on the oxygen sensitivity of this process (e.g. Kalvelage et al, 2011; Bristow et al, 2016 etc) and the potential overlap with nitrite oxidation making this even more of an exciting topic.

We thank the reviewer for the helpful suggestion, and agree that this makes this an even more exciting topic. While we do discuss nitrate reduction in more detail later in the paper, introducing this idea here makes sense, and ties into our ending comment about precluding nitrite's use as an oxidant. We now state on lines 79-81 that "SCM-based DO production can also overlap with nitrite supply via nitrate reduction—which occurs at higher DO levels than previously thought, expanding the depth range over which nitrate reduction and nitrite oxidation may interact." We reference Kalvelage et al. 2011 and Bristow et al. 2016.

Line 88 to 90: It is not clear to me what omic data the authors are referring to in the literature cited as evidence for alternative electron donors and acceptors for NOB? What about data beyond omics that suggests a role for anaerobic nitrite oxidation, a more thorough introduction seems warranted here.

We apologize for a citation mistake here, as this previously should have been citation 23 (Sun et al. 2019) rather than 13 (the citation numbering is now different). Both the Sun et al. 2019 and 2021 papers discuss their omic (and biogeochemical) data at some length, and mention a variety of different scenarios that might support their findings. For example, Sun et al. 2019 state that "NOB might survive in ODZs by alternative anaerobic metabolisms and turn on the conventional nitrite oxidation pathway when minute amounts of O₂ become available." Sun et

al. 2021 devote a section of the Discussion to nitrite oxidation in the ODZ, ending with “NOB may also use yet to be discovered metabolisms to survive in anoxic waters.” So we think that with the fix to the citation here, we accurately summarize the papers, and apologize for the mistake.

That said, the broader point about other evidence for anaerobic nitrite oxidation is a really good one. We made this sentence more general by deleting “Omic data further indicate that,” and by adding a citation to a recent paper by Babbin et al. that presents evidence for anaerobic oxidation.

However, this does tie into the Reviewer’s comment directly below, as previous studies don’t have DO measurements within sample bottles, so whether incubations are truly anaerobic—or have nM levels of oxygen present—is unclear. We also note that the issue of which alternative oxidants are actually used came up in our response to previous reviews. The Sun et al. papers clearly back off on the idea of iodate as an alternative oxidant. So while it does seem that nitrite may be oxidized anaerobically, it is still unclear exactly how. Given all of this, we feel that the best approach at this point in time is to keep this to one sentence that notes the evidence and possibility of anaerobic oxidation.

Line 92 to 94: while I 100% agree that variations in genomic content is a potential explanation here, high insitu rates under low DO (putting a number here would be helpful for the reader) could also surely be explained at least to some extent by oxygen contamination – which surely only strengthens your study as you measured oxygen in your incubations. I know that you come to this point in your discussion but it also seems relevant here.

Adding a number here is a great idea and we added “(>100 nmol L⁻¹ d⁻¹)” along with several references that have shown this. We also agree that having in situ oxygen measurements is a strength of our work, and we thank the reviewer for pointing this out. We added additional text (lines 98-102) stating that “High DO affinity among some NOB also implies sensitivity to any DO introduced to incubations, and may partly explain some of the elevated rates reported in OMZs. However, DO is rarely measured directly during nitrite oxidation rate measurements, limiting our understanding of NOB affinity for DO and their activity within OMZs.”

Results / Discussion: as currently presented there is a lot of discussion in the results section (e.g. 178 to 190, 230 to 238 etc) and the discussion more of a summary in places, maybe this could become a single results / discussion section? (some restructuring would be needed to make this work)

We agree that some Discussion has crept into the Results here, but the text noted by the reviewer was included here as a direct result of the first round of reviews. We feel that it is a better fit here, as the reviewers clearly highlighted the need to compare our results with the literature. While there are some summarized results in the Discussion, this is actually quite limited, and is important for guiding readers through the implications (and caveats) of our integrated findings. Finally, the journal format is to separate Results and Discussion. So in sum, we feel that the current format is best as-is given the need to integrate our results and compare them with the literature.

Line 151-152: I value that the authors have included the starting DO concentrations in their incubations, and then later for OCR (line 175) state that these rates are likely potential due to this, why is there not a similar statement for nitrite oxidation rates? Here seems like a good place to do it, to be upfront.

This is a good point, although later in the paragraph we mention that this will be ‘discussed below’ (which we do when presenting the OCR results). We added and modified the latter half of that sentence (now lines 159-163) to state that “Elevated nitrite oxidation... should be considered potential rates and could have a number of possible explanation discussed below.”

Line 188: if you want to include the value from Namibia here, I think you need to be clear that it is from shelf waters and not offshore deep waters like your own work and that for the values you present from Kalvelage et al, 2015.

We modified this sentence to note that this is “in the SCM in Namibian shelf waters.”

Line 220 and 221: what is the basis for this being the cutoff for potential rates?

We changed this to “>100s of nM” and reference Table 1 for the starting concentrations.

Line 304: why are you only correlating your DNA / RNA with the natural abundance data, surely your rates are more appropriate with respect to timescale.

We agree that examining relationships between DNA/RNA and rates makes sense, and we now provide the statistically significant relationships observed at the AMZ stations (line 301), which we feel strengthens our points here.

Line 308 to 342: this is super interesting! Why is a plot of the individual ASVs not included in the manuscript? This really helps bring your ideas together. It would also be valuable here to contrast your metagenomic findings to that of other OMZ studies such as Sun et al, 2019 in more detail (e.g. terminal oxidases, nitrate reductase etc), apart from one sentence regarding chlorite dismutase, this is very much a results section and it would be valuable for the reader for you to compare / contrast your findings to work to date.

We thank the reviewer for the suggestion about the ASVs, and now include plots of two ASVs that are important in the upper water column and AMZ as a Supplementary Figure 5.

Re the metagenomic data, we could obviously include much more detail, but as we mentioned in our previous response, we use the metagenomic data here to support our main findings. We will be analyzing these data in various ways in subsequent publications, and to go through all of the relevant genes in comparison to early work would be overkill, as we present information for 6 genes and 2 bacterial groups—all of which can and should be discussed at length in more omic-focused publications. But we do agree that some comparison is warranted here, and so we included a sentence about our top-line finding stating that “*nxr* and all *Nitrospina* genes were most abundant in the SCM, consistent with a range of omic data showing maximal *Nitrospina* gene or enzyme abundances in the upper portion of AMZs” and referencing a range of relevant studies.

Line 384: an additional reference here would be Larsen et al, 2016 In situ quantification of ultra-low O₂ concentrations in oxygen minimum zones: application of novel optodes. L&O Methods 14 (784-800) (see Figure 6 and text)

Thank you for the suggestion; we added Larsen et al. as a reference here as well as earlier in the paragraph.

Line 396: please clarify what you mean by dominant here.

Here we are referring to >50% of OCR and have now defined this.

Line 440: what experiments were conducted in which year?

We now specify this later in the Methods when we describe our measurements.

Line 463 to 464: is the 10nM detection limit taken from the Lehner et al paper? As I understand it that paper is discussing completely different optode chemistries to that of the commercially available trace optodes available from Pyroscience. What spots / chemistry did you use for the oxygen manipulation experiments?

The 10 nM limit is from the manufacturer. The reference to Lehner et al. here was meant to support the overall idea of using DO sensor spots for these measurements—not the specific spots—but this is obviously confusing. We removed the reference here.

Line 471: this sentence is not at all clear to me 'within 10nM of zero'?

Reviewer 3 also commented on this and made a helpful suggestion to change this, which we have followed; please see our response to Reviewer 3 at the very end.

Line 473 to 475: this seems out of place and no justification is provided

We removed this sentence for this manuscript. We originally had some more text on this but removed it prior to original submission, and will likely include more in a future paper.

Figure 3: in the caption I think you need to link symbol to location in the water column for panel C e.g. Station 2 (SCM (closed purple) and OMZ edge (open purple)

We now specify this in the figure caption. We also changed the symbols so that they consistently correspond with different depth regions, which helps further clarify the higher relative rates of nitrite oxidation in the SCM and SNM.

Figure 4: would it be possible to reorganize the panels so that OMZ edge, SCM and SNM aligned vertically, this would make it easier to compare.

Another great suggestion and we modified the order of the figure panels so that they are vertically organized by depth region rather than numerically by experiment.

Table 1: it would be valuable here to include the errors on your K_m / V_{max} numbers

We added standard errors to the Table. In doing this, we decided it was better to not include any numerical information for the estimates that were insignificant.

Table 2: DO, I assume this is from the SBE 43 on the CTD? If yes, what is the detection limit of the sensor, this is needed so that readers do not assume that oxygen was measurable at all depths sampled e.g. 0.3 μM at the SNM. More information is needed here regarding the functional genes e.g. nitrite oxidoreductase (nxrB?). All of these comments are also relevant for Table S1.

We apologize for not specifying this, but these values were measured using sensor spots in bottles immediately after sample collection and prior to any manipulation. We now specify this. We also added (*n_{xr}*) to the Table and provided more detail for all of the columns. Similar changes were also applied to Supplementary Table 1.

Reviewer #2 (Remarks to the Author):

Beman et al., Constraining the contribution.....

The authors have answered very appropriately to the questions asked from the three reviewers of the first version. I thus feel that the manuscript now is in a good shape. However, I still recommend a few minor changes.

Intro: A small typo: refs 10 and 11 should be superscript.

We thank the reviewer catching this, although the guide to authors suggests that references should be formatted like this when a superscript would introduce ambiguity. We think this applies to the first case here, whereas the second case is less clear. We kept “ref. 10” as-is, but modified ref. 11 to superscript; in any case, this would be addressed by the professionals in production in order to fit the journal style.

Figure 2: It should be mentioned in the figure legend (and also more specifically in the main text) that the OCR data are potential rates. Most would feel it strange that the authors find positive OCR values in the anoxic layers!

This is a good point and we added a sentence to the figure caption in case the readers miss this in the main text.

Line 181: I have had a look on the photosynthesis data of ref. 15, and they fall far below the values given here. At an in situ light level the photosynthesis rate at noon is below 5 and 10 nmol L⁻¹ h⁻¹ in the ETNP and ETSP, respectively, so rates are below 50 and 100 per day, respectively. Respiration rates are variable but more realistic. So it would be better not to write that photosynthesis and respiration rates are in the same range.

On the line number identified by the reviewer, these values refer only to the OCR measurements reported in ref. 15—so obviously this wasn’t written very clearly. We deleted “production and” from the first part of the sentence, and added “OCR” before “rates” to clarify this.

However, earlier in the manuscript, we do refer to the DO production rates reported in ref. 15, and presented the full range of values from their experiments and calculations. These include elevated light levels, and we agree that the in situ production values are typically much lower, so we changed this to “up to ca. 100 of nM-O₂ d⁻¹ occur in the SCM.”

Line 215-217: It is actually not very clear that nitrite reduction is decreased at low O₂ in some of the experiments, and I thus do not think that the authors should use the word “steep”

We apologize for not being specific here, but this refers to the OCR values discussed in the previous sentence of our manuscript, not to nitrite oxidation (the reviewer writes ‘nitrite reduction’ above, but we obviously assume s/he intended to write ‘nitrite oxidation’). We agree

that nitrite oxidation does not show steep declines, which is a key point of the paper when contrasted with OCR. To clarify this sentence we added “in OCR” after “These declines...”

Line 250 - 264: This is too speculative, and I suggest the section modified. It is clear that nitrite oxidation will never account for all oxygen consumption, and any extrapolation of consumption rates down in O₂ concentration to obtain 100% consumption by nitrite oxidation has no meaning! But that nitrite oxidation becomes progressively more important by decreasing O₂ is correct!

We included this section previously assuming that some reviewers/readers may wish to see this calculation. We agree that pushing out to 100% is probably too speculative, and so we removed these calculations from the manuscript.

Of the remaining text, information about variations across experiments (which doesn't include calculations, just description of our results) was moved into the previous paragraph (now lines 252-258). We think that this point—as well as the fact that we wouldn't expect nitrite oxidation to be as significant on the OMZ edge—is important to make. We moved the statistical information for the relationships reported in Figure 3 to the concluding sentence of the previous paragraph (line 261).

Lines 270-271: “NOB are therefore effective at scavenging DO (Figs. 3 and 4; ref. 15), and the concentration range over which this occurs in our experiments matches model predictions (0.2-1 μM DO)” – I do not like this sentence, as do scavenging occurs throughout all O₂ concentrations, but at varying importance – so rewrite.

This is a really good point and we agree fully with the reviewer. We have changed the second part of the sentence to state that “the concentration range over which they become increasingly important matches model predictions.”

Line 403-5: Nitrite oxidation is also a respiration, so please write heterotrophic respiration in contrast to OCR by nitrite oxidation.,

Good point, and we added ‘heterotrophic’ to this sentence to be more specific.

Reviewer #3 (Remarks to the Author):

Beman et al. have addressed most of the comments of the reviewers. There are still a few things that could be improved.

Reviewers 1 and 2 both wanted to know if there were lags or non-linear behavior during the course of the incubations. The authors added discussion of the trends in the text but it is still unclear. This point about linearity is important to make clear so the authors should present in figures at least some of the timecourses that they have measured – moles of NO₂- oxidized vs. time – in the supplemental would be fine.

One reviewer previously inquired about nonlinearity over time for the nitrite oxidation rates, while several asked about nonlinearity in OCR. Given our focus, and the time and expense of the ¹⁵N measurements, we noted in our previous response to reviewers that it was more useful to examine nitrite oxidation at additional depths and DO levels, rather than over time. Re OCR, we included additional information on nonlinearity in oxygen consumption over time—including

the multiple DO sampling time points for profile measurements, as well as the continuous measurements in the experiments. We described and analyzed these data following the comments of the reviewers, and followed Tiano et al. in not including any oxygen time courses. But given the interest in this, we included example oxygen time courses in the supplemental to illustrate the nonlinearity in DO consumption that occurred at <235 nM DO in some of the replicates (as described in the main text).

Also related to this discussion of linearity in the timecourses:

Lines 482-485 “slopes ranging from 1.06-1.19” What are the units for these values? These slopes are not a large enough range to represent the range in rates presented in Fig. 2.

These slopes refer to the statistical relationships between OCR rates calculated at the two different measurement time points, not the slope of DO over time (which doesn't compute properly, as the reviewer notes). To clarify this, we included the slope information with the other stats (r^2 and P values), and also added additional text noting the two measurement time periods. This sentence now reads: “The majority of OCR rate values calculated at these 10-14 hour and 20-24 hour measurement time points were highly correlated with each other ($r^2 = 0.968-0.995$; slopes=1.06-1.19; all $P < 0.0001$ across different stations), indicating that OCR did not accelerate or decrease substantially over the course of the incubations.”

Lines 485-487 “The only exceptions were three sampling depths...showing nonlinearity, and so we report data from measurements conducted at similar depths...the previous day.” Did all the rest of the “profile” rate measurements occur on a single day? Why was this so important? Does this mean the only nonlinear timecourses observed were from the second day (I say “second” only because the authors refer to the “previous” day, not because I understand what the authors mean here)? And these nonlinear timecourses were then...disregarded?

We apologize for the confusion here. These are a duplicate set of measurements made on the previous day without corresponding nitrite oxidation measurements. Given comments by the reviewers on our previous manuscript, we think that it is much more prudent to use (linear) rates measured the previous day, rather than these three OCR rates showing nonlinearity. We again note these are the only exceptions to linearity in the profiles, and also that other duplicate measurements all closely agree. We agree that referring to the ‘previous day’ is confusing, and so changed this to ‘previous 24 hour period.’ Overall we added additional text to this sentence to clarify that these are “duplicate OCR measurements” conducted “during the previous 24 hour period (when nitrite oxidation rates were not measured in tandem).”

Reviewer 1 suggested adding the in situ DO concentrations, specifically in the context of the detection limit of the Seabird SBE-43. The authors added some of this information in Table 1 however it could be further clarified:

1) It should be labeled “in situ DO measured by SBE-43” in Table 1 otherwise it is unclear if “starting DO” is the DO measured in the incubation bottle after collection.

Reviewer 1 also commented on this (see above) and we are sorry for not specifying this. As mentioned above, these DO values were measured using sensor spots. We now specify this below the Table.

2) The authors should include the in situ DO concentrations for the “profile” incubations as well.

We thank the reviewer for the suggestion, as this definitely helps clarify our sampling, and we added an additional Supplementary Table 2 with this information.

3) The SBE-43 has a published accuracy of +/-2% of saturation. The authors are reporting DO concentrations well below this. If they did additional calibrations to resolve lower DO concentrations with the SBE-43 they should explain what they did. Otherwise, "below detection" should be stated.

Please see our notes above; we now specify that these values are not from the SBE-43, but from sensor spots.

Also related to detection limits:

Lines 471-471 "our lowest values were consistently within 10nM of 0" This suggests that the values <10nM are significant. Instead the authors should state "our lowest values were consistently below the detection limit of the FireSting"

This is a good point of clarification that was also raised by Reviewer 1, and we have revised the sentence following the recommendation of Reviewer 3 to state that "our lowest values measured in experiments were below the detection limit (10 nM) of the FireSting"

REVIEWERS' COMMENTS

Reviewer #1 (Remarks to the Author):

I would like to thank the authors for more than adequately responding to mine and the other reviewers' comments (Figure S5 is particularly exciting, and I look forward to seeing the future molecular papers from this work). I just have a few very minor comments / suggestions

Line 94: please include what you mean by low DO, so ' $< X \mu\text{M}$ ' for where nitrite oxidation rates of this magnitude have been observed to date.

Line 384: I think the Larsen reference (no. 46) belongs with the next sentence on oxygen intrusions and not with this sentence on Prochlorococcus abundances (these were not looked at in this reference).

Table 1 caption and onwards: in a number of the figure captions (including in the supplement) you have added Firesting in brackets, I would update this to Firesting, Pyroscience throughout.

Supplementary Table 2: you state that values are below the detection limit of the SBE 43, but never tell us what that detection limit is, please add.

I look forward to seeing this work in print and citing it in the future
-Laura Bristow

We thank Reviewer 1 for her final helpful comments on our revised manuscript, which we feel have further improved the manuscript. Below we provide detailed responses these comments, with our **responses in blue** text.

Reviewer #1 (Remarks to the Author):

I would like to thank the authors for more than adequately responding to mine and the other reviewers' comments (Figure S5 is particularly exciting, and I look forward to seeing the future molecular papers from this work). I just have a few very minor comments / suggestions

Line 94: please include what you mean by low DO, so '< X μM ' for where nitrite oxidation rates of this magnitude have been observed to date.

This is a good suggestion. We moved the numerical information for nitrite oxidation rates earlier in the sentence to avoid confusion, and added the relevant DO concentration (<11 μM) from the cited studies (refs. 6-9).

Line 384: I think the Larsen reference (no. 46) belongs with the next sentence on oxygen intrusions and not with this sentence on Prochlorococcus abundances (these were not looked at in this reference).

We agree that this is a better location for this reference. We made this sentence more general (referring to both the Tiano and Larsen studies) and added the relevant references at the end of the sentence.

Table 1 caption and onwards: in a number of the figure captions (including in the supplement) you have added Firesting in brackets, I would update this to Firesting, Pyroscience throughout.

We updated all of these instances to "Firesting, Pyroscience"

Supplementary Table 2: you state that values are below the detection limit of the SBE 43, but never tell us what that detection limit is, please add.

We apologize for this iversight and added the detection limit of 1 μM to the note at the end of the table.

I look forward to seeing this work in print and citing it in the future
-Laura Bristow